# MYC modulates TOP2A diffusion to promote substrate detection and activity

Donald P. Cameron [1,8,10], Kathryn Jackson[1,10], Alessia Loffreda [2], Carl Möller [3], Vladislav Kuzin[1], Matteo Mazzocca [2,9], Evanthia Iliopoulou[1], Hallgerdur Kolbeinsdottir[1], Andrej Paluda [4], Evgeniya Pavlova[3], Bea Jagodic [1], Brian Saidel Lopez Duran[5], Valérie Lamour [5,6], Fredrik Westerlund [3], Davide Mazza [2,7] ✉ & Laura Baranello [1] ✉

Topoisomerases alleviate DNA supercoiling by cleaving and resealing DNA strands. Previously, we showed that the oncoprotein MYC recruits and stimulates topoisomerases to remove DNA entanglements generated by oncogenic transcription. Understanding this mechanism may suggest methods to inhibit MYC-driven topoisomerase activation, targeting tumor-specific transcription. Here, we demonstrate that the essential topoisomerase TOP2A in human cells exists in a dynamic equilibrium between sequestration in the nucleolus, substrate searching in transcription hubs, and active engagement on chromatin. This equilibrium is highly responsive to changes in DNA topology, allowing cells to regulate TOP2A levels. Using single molecule tracking, here we show that MYC accelerates TOP2A diffusion in cells. We explain this phenotype by demonstrating that MYC limits TOP2A self-interaction in vitro, while decreasing the size of TOP2A complexes in cells. By increasing TOP2A diffusion, MYC promotes substrate binding and increases TOP2A engagement on chromatin genome-wide, revealing the mechanism underlying MYC stimulation of TOP2A activity.

MYC functions as a global activator of transcription, increasing the amplitude of transcriptional bursts and boosting promoter output[1]. Transcription also generates DNA supercoiling downstream and upstream of an elongating RNA Polymerase (RNAP)[2]. As MYC increases transcription rates, it drives more torsional stress into the DNA template, which, if not resolved, can slow down transcription and hinder MYC-driven overexpression[3]. Thus, MYC's ability to increase transcriptional output depends on efficient relief of topological problems.

Topoisomerases relax DNA supercoils and decatenate intertwined DNA by cutting and resealing DNA strands. Topoisomerase 1 (TOP1) creates a transient single-strand break, allowing rotation around the unbroken strand to release DNA twist. Topoisomerase 2 (TOP2) generates a transient double-stranded break to pass one intact DNA duplex through the break, thus resolving supercoils or DNA catenanes. Both enzymes facilitate transcription by relieving topological stress. Although topoisomerases exhibit limited sequence specificity, their activity is directed by recognition of DNA topology or interaction with partner proteins[4]. Previously, we demonstrated that MYC can recruit and stimulate both TOP1 and TOP2A, localizing them to sites of high transcription, thereby promoting increased transcription rates[5]. However, the precise mechanism by which MYC mediates topoisomerase recruitment and stimulation remains unclear.

[1]Department of Cell and Molecular Biology, Karolinska Institutet, Stockholm, Sweden. [2]Experimental Imaging Center, IRCCS Ospedale San Raffaele, Milano, Italy. [3]Department of Life Sciences, Chalmers University of Technology, Göteborg, Sweden. [4]Institute for Quantitative Biosciences, University of Tokyo, Tokyo, Japan. [5]Université de Strasbourg, CNRS, INSERM, Institut de Génétique et de Biologie Moléculaire et Cellulaire (IGBMC), Strasbourg, France. [6]Hôpitaux Universitaires de Strasbourg, Strasbourg, France. [7]Vita-Salute San Raffaele University, Milano, Italy. [8]Present address: Institute for Quantitative Biosciences, University of Tokyo, Tokyo, Japan. [9]Present address: Department of Biological Engineering, Massachusetts Institute of Technology, MA Boston, USA. [10]These authors contributed equally: Donald P. Cameron, Kathryn Jackson. ✉e-mail: mazza.davide@unisr.it; laura.baranello@ki.se

TOP2A, the isoform associated with proliferating cells, has a disordered carboxyl-terminal domain (CTD) that mediates both chromatin targeting and catalytic function[6,7]. Notably, TOP2A can form CTD-dependent phase-separated condensates in vitro, which partition with plasmid DNA at high concentrations[8]. These condensates reflect a transition in the activity of TOP2A from relaxation and decatenation to catenation, suggesting that TOP2A phase-separates on chromatin during mitosis to induce chromatid catenation, which is essential for chromosome condensation[9]. During interphase, transcription condensates containing MYC form at active sites, including super-enhancers, to drive transcription activation[10]. Because these condensates are often associated with elevated transcription[10,11], they are likely to include TOP2A to relieve topological stress. Although TOP2A condensates that favor catenation may be beneficial for mitotic condensation, this action would hinder the removal of topological stress generated by transcription in interphase if local levels of TOP2A reach concentrations associated with condensate formation. It remains unclear whether TOP2A supports transcription via association with the interphase condensates.

In this study, we demonstrate that TOP2A maintains a dynamic equilibrium between nucleolar and transcription condensates in human cells. This equilibrium is maintained by the integrity of these condensates and is highly responsive to changes in DNA supercoiling. We found that MYC accelerates TOP2A diffusion in cells. In line with biophysical principles, MYC reduces the size of the TOP2A condensates in vitro and decreases the size of TOP2A-containing complexes in cells, providing a rationale for the enhanced diffusion. By increasing TOP2A diffusion, MYC favors TOP2A substrate detection and promotes TOP2A catalytic engagement on chromatin genome-wide, revealing the mechanism for MYC stimulation of TOP2A activity.

## Results

### TOP2A exists in an equilibrium between nucleolar and transcription condensates

We first examined TOP2A localization in interphase HCT116 cells, a colorectal cancer cell line previously used for studies of topoisomerases[5,12]. We applied stimulated emission depletion (STED) microscopy[13] to achieve sub-diffractive optical resolution of TOP2A. Co-staining of nucleolar substructures—the fibrillar center (RNAPI), the dense fibrillar compartment (Fibrillarin), and the granular component (Nucleophosmin)—revealed that TOP2A was enriched across all three compartments (Fig. 1A, Supplementary Fig. 1A). Since rRNA transcription occurs specifically at the boundary of the fibrillar center[14], this indicates that the nucleolar TOP2A enrichment is not specific to its known site of activity during ribosomal RNA (rRNA) transcription[15]. Following treatment with buffer to remove soluble proteins[16], TOP2A enrichment in the nucleolus persisted, but was lost along with the nucleolar protein fibrillarin upon further treatment with RNase A (Fig. 1B, Supplementary Fig. 1B), suggesting its association with nucleolar structures was maintained by ribosomal RNA. Outside of the nucleolus, STED imaging also revealed heterogeneity of TOP2A, showing enrichment in nucleoplasmic puncta (Fig. 1A, Supplementary Fig. 1C). We predicted that these puncta correspond to transcription condensates, given TOP2A's involvement in transcriptional regulation[17]. Indeed, co-staining with MED4, a component of the Mediator complex enriched in transcription condensates[18], showed strong colocalization with TOP2A (Fig. 1C, Supplementary Fig. 1C), while treatment with 1,6-hexanediol, shown to rapidly dissolve Mediator-containing transcriptional condensates[18], weakened TOP2A puncta intensity relative to the surrounding area (Fig. 1D, E, Supplementary Fig. 1D). The association between the two proteins was further supported by overlapping ChIP-seq signals of TOP2A and MED26—another subunit of Mediator—from previously published datasets[5,19] at MED26 peaks (Supplementary Fig. 1E). These data suggest that TOP2A puncta may partially correspond to Mediator-containing transcription

condensates, although they could also include other sub-nucleoplasmic structures.

Previous studies have demonstrated that TOP2A exists in an equilibrium between the nucleolus and the nucleoplasm[20,21]. This equilibrium persists as the TOP2A concentration fluctuates across cell cycle phases (Supplementary Fig. 1F, G). Given the enrichment of TOP2A in the nucleolus and nucleoplasmic transcription condensates, we propose that the integrity of these compartments supports this equilibrium. To challenge this hypothesis, we first observed how TOP2A distribution changes after these two compartments are targeted by drug treatment. Inhibition of RNAPI elongation with 5 nM actinomycin D for 1 h reduced ribosomal RNA expression (Supplementary Fig. 1H) and induced nucleolar-to-nucleoplasmic translocation of TOP2A (Fig. 1F, Supplementary Fig. 1I), as previously reported with other RNAPI transcription inhibitors[21]. Given that this treatment did not extensively affect the nucleolar structure under our experimental conditions, as observed by fibrillarin staining (Fig. 1F), it suggests that TOP2A is largely retained in the nucleolus via rRNA-mediated nucleolar interactions. This was previously observed with rat TOP2A, where the CTD confers both nucleolar localization and sensitivity of TOP2A activity to RNA inhibition[22]. Conversely, treatment with 1,6-hexanediol caused TOP2A to shift from the nucleoplasm into the nucleolus within 5 min (Supplementary Fig. 1D). While 1,6-hexanediol can dissolve many condensates, including Mediator-containing transcriptional condensates[23] (Supplementary Fig. 1D), by disrupting weak hydrophobic interactions, it can favor condensation of other compartments through hydrophilic interactions[24]. Indeed, other phase-separated proteins that are maintained through hydrophilic interactions are not dissolved by 1,6-hexanediol treatment[25]. This could include proteins, such as nucleophosmin, which is enriched in the nucleolus by its positively and negatively charged disordered domain[26]. This may explain why TOP2A enrichment in the nucleolus is not disrupted by 1,6-hexanediol treatment. Since the dysregulation of either the nucleolus or nucleoplasmic transcription condensates causes the rapid translocation of TOP2A from one region to the other, the data suggest the equilibrium is at least in part maintained by these two structures.

The presence of a dynamic equilibrium might allow for rapid TOP2A translocation upon changes in DNA topology. We predicted that the TOP2A equilibrium enables the cell to rapidly respond to changes in the supercoiling burden. If this is the case, increasing the level of supercoiling in the genome should provoke a redistribution of TOP2A from the nucleolus to the nucleoplasm. We tested this hypothesis in two ways. Firstly, because TOP1 and TOP2 can partially compensate for each other's function, degradation of one enzyme is expected to trigger a compensatory increase in supercoiling, thereby enhancing engagement of the other topoisomerase on the DNA[27]. We therefore acutely degraded TOP1 by auxin[28]. Secondly, we treated cells with ICRF-193, which blocks TOP2A recycling after DNA religation[29,30]. Both treatments have been reported to increase genomic supercoiling by psoralen binding assay[31]. In both tested conditions, we observed a rapid translocation of TOP2A from the nucleolus to the nucleoplasm and nucleolar periphery (Fig. 1F, G, Supplementary Fig. 2A, B) without affecting total nuclear TOP2A concentration (Supplementary Fig. 2C, D). To test whether the translocated TOP2A was actively engaged on the DNA, we treated cells with a low dose (1 μM) of etoposide for 60 min. Etoposide traps covalently engaged TOP2 on the DNA, causing DNA damage, which we can visualize with γH2AX staining. We observed an increase in DNA damage only after TOP1 degradation by auxin and etoposide treatment, suggesting that the increased nucleoplasmic fraction of TOP2A after TOP1 degradation correlates with increased TOP2 activity (Fig. 1H, Supplementary Fig. 2E). These γH2AX puncta were enriched within the nucleoplasm relative to the nucleoli, indicating that the increased activity is spread across the genome rather than being specific to the ribosomal DNA loci

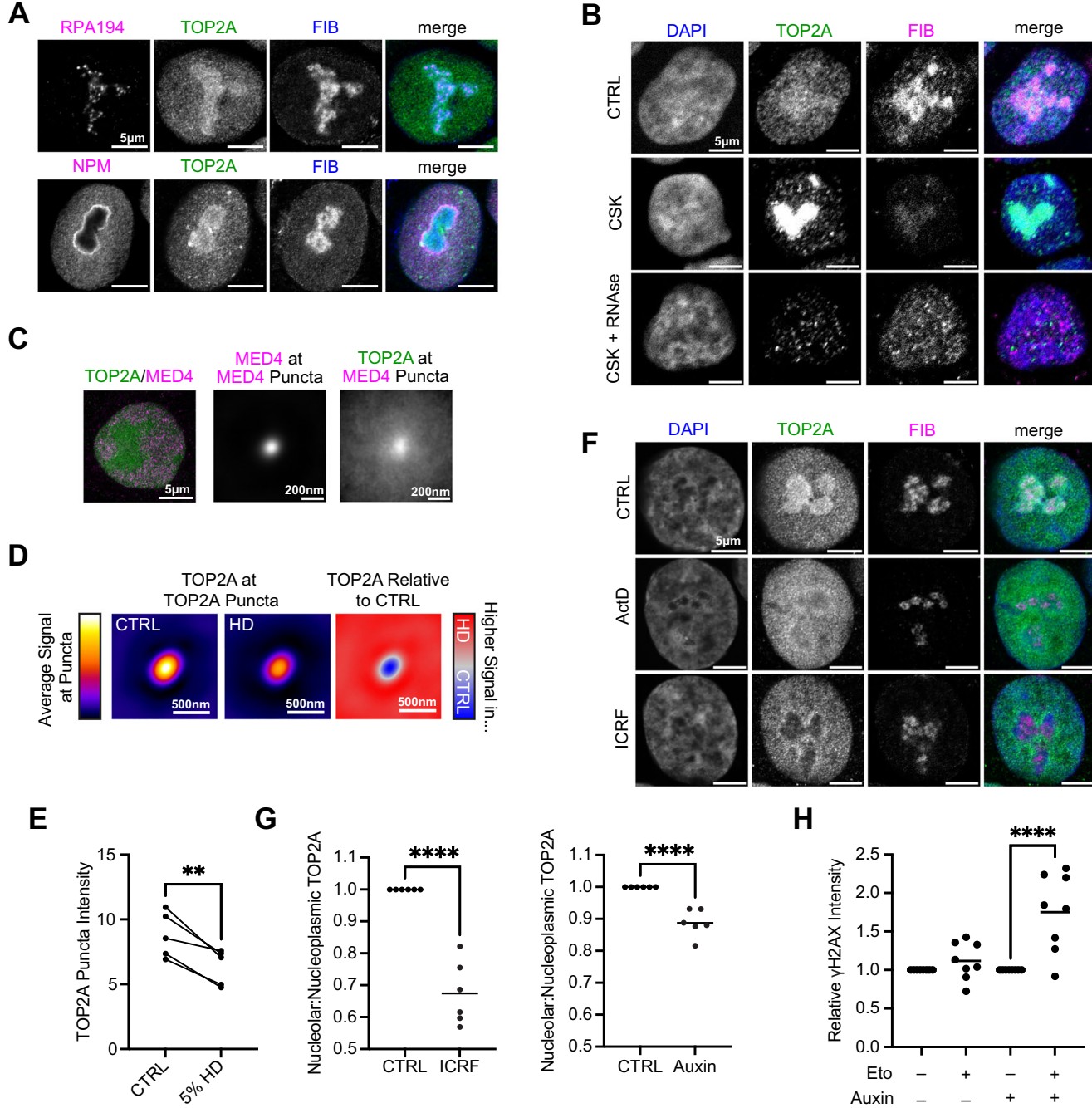

**Fig. 1 | TOP2A exists in an equilibrium between nucleolar and transcription condensates. A** STED images of HCT116 cells immunostained for TOP2A, RNAPI subunit RPA194, and nucleolar proteins Fibrillarin (FIB) and Nucleophosmin (NPM). Two independent experiments were performed. Scale bar = 5 μm. **B** Confocal images of HCT116 cells treated with CSK buffer to remove soluble proteins +/- RNAse immunostained for TOP2A and FIB, and DNA stained with DAPI. Four independent experiments were performed. Replicate 1 is shown. Scale bar = 5 μm. **C** Representative STED image of HCT116 cell (left) immunostained for TOP2A and Mediator component MED4. Average MED4 (middle) and TOP2A (right) signal at 8240 MED4 puncta from 30 cells from three independent experiments. Scale bar as indicated. **D** Average TOP2A signal at TOP2A puncta from STED images in control (CTRL, left) or upon 5% 1,6-hexanediol treatment (center) and difference between both conditions (right). Signal was averaged from five independent experiments, 4100–4180 puncta per condition. **E** Average TOP2A puncta enrichment across independent experiments from data in (**d**). **p = 0.0072 (Two-tailed paired t-test). **F** STED images of HCT116 cells treated with 5 μM TOP2 inhibitor ICRF-193 for

30 min or 5 nM Actinomycin D (ActD) for 1 h to specifically inhibit RNAPI, immunostained for TOP2A and FIB, and DNA stained with DAPI. Two independent experiments were performed. Scale bar = 5 μm. **G** Quantitation of nucleolar:nucleoplasmic ratio of TOP2A signal normalized to untreated control after treating HCT116 cells with 5 μM ICRF-193 for 15 min (left) or HCT116^TOP1-AID cells (right) with 500 μM auxin for 3 h, to degrade TOP1. The plot of ICRF-193 treatment shows the average of six independent experiments, with each point representing per experiment means, and the plot of auxin treatment shows the average of six independent experiments with each point representing per experiment medians. 550–889 cells per condition were measured. ****p < 0.0001 (unpaired two-tailed t-test). **H** Nuclear γH2AX intensity of HCT116^TOP1-AID cells ± 500 μM auxin for 3 h ± 1 μM etoposide (Eto) treatment for 1 h, normalized to respective "– Eto" conditions. The plot shows the average of eight independent experiments, with each point representing per experiment means. 1020–1378 cells per condition were measured. ****p < 0.0001, ns p = 0.959 (Ordinary one-way ANOVA, Šidák correction).

(Supplementary Fig. 2F, G). It is also possible that part of the γH2AX signal is associated with the pre-existing nucleoplasmic TOP2A, trying to compensate for the absence of TOP1.

Overall, these findings suggest that TOP2A localization is regulated by its interaction with both nucleolar and transcription condensates, existing in an equilibrium between compartments that is sensitive to both their dysregulation and DNA supercoiling. Notably, TOP2B distribution across the nucleus was much more homogeneous, as compared to TOP2A, although ICRF-193 also caused a slight nucleolus-to-nucleoplasm relocalization (Supplementary Fig. 2H, I), suggesting it may also be responsive to similar stimuli to TOP2A, although the TOP2A response was more pronounced.

## The dynamics of TOP2A in the nucleus can be described by three states

To better characterize the dynamic equilibrium of TOP2A between its associated compartments, we tagged endogenous TOP2A with a Halo-tag and used a custom Highly Inclined Laminated Optical sheet (HILO) imaging platform[32] to perform Single Molecule Tracking (SMT) in live HCT116 cells. This approach allowed us to measure the location and motion of single fluorescently labeled TOP2A molecules over time (Fig. 2A). Single TOP2A molecules exhibited a spectrum of diffusion coefficients, peaking at very low diffusion coefficients (corresponding to immobilized molecules) and with a second peak at ~1 $\mu m^2$/s (corresponding to molecules quickly diffusing through the nucleus). Between these two peaks, an intermediate regime, corresponding to a shoulder in the spectra at ~0.1 $\mu m^2$/s was visible (Fig. 2B, Supplementary Movie 1). Notably, TOP2A mobility was substantially reduced when trapped on DNA by ICRF-193 treatment[33] (Fig. 2B, Supplementary Movie 2).

To assign individual track segments to different diffusing states, we used a Hidden Markov Model approach (variational Bayes single particle tracking, vbSPT)[34] and characterized nuclear TOP2A diffusion into three distinct populations. The slowest population had diffusion coefficients below 0.1 $\mu m^2$/s (Fig. 2C), comparable to the mobility of histones and chromatin-bound proteins[35]. We refer to this group as the bound fraction. The other two populations, which represent TOP2A molecules diffusing in the nucleus, are referred to as slow and fast fractions based on their diffusion rates (Fig. 2C). Treatment with ICRF-193 caused a marked increase in the bound population (Fig. 2C). As it takes over an order of magnitude longer for strand passage by TOP2A to occur than the framerate of our imaging (300 ms for strand passage[36,37] compared with 10 ms between frames), and since DNA-bound TOP2A exhibits diffusion rates far lower than TOP2A in the fast and slow fraction, we conclude that this fraction includes TOP2A covalently engaged on the DNA. However, this may also include TOP2A that is interacting with chromatin without catalytic engagement. TOP2A in the fast and slow states is not precluded from becoming covalently engaged on the DNA, given the observed transitions between the fast and slow fractions as well as between the slow and bound fractions (see arrows in Fig. 2C). It is notable that fast-to-slow and slow-to-bound transitions are substantially more frequent than fast-bound transitions, suggesting the TOP2A molecules must pass through the slow state before becoming chromatin-bound.

Using a GFP-Nucleolin marker to separately analyze TOP2A diffusion in the nucleolus and the nucleoplasm, we observed that diffusion speed was reduced by more than 50% in the nucleolus relative to the nucleoplasm, with a rate of 0.61 $\mu m^2$/s in the nucleoplasm as compared to 0.28 $\mu m^2$/s in the nucleolus. In addition, the nucleolus predominantly contained slow and bound TOP2A, while the nucleoplasm contained more equal amounts of all three populations (Fig. 2D, E). This indicates that TOP2A diffusion is more constrained within the nucleolus.

The fast and slow fractions are further distinguishable by their diffusional anisotropy, with the nucleoplasmic slow fraction exhibiting anisotropy values greater than 1 (Fig. 2F), indicative of protein retention within discrete compartments. Nucleoplasmic tracks with average anisotropy greater than 1 were limited to regions of about 150 nm in size (Fig. 2G), consistent with both the approximate size of the MED4 puncta (Fig. 1C) and the reported size of a transcriptional condensate[38]. These features suggest that the slow TOP2A fraction in the nucleoplasm likely contains molecules performing compact target searching in transcription condensates[39]. Importantly, previous studies have shown that transcription condensates associate with gene loci only transiently[11]. Thus, if the slow TOP2A fraction represents TOP2A localized in transcriptional condensates and the bound fraction corresponds to TOP2A interacting directly with chromatin, we would expect the slow fraction to border the bound fraction, reflecting transient spatially proximal interactions between transcription condensates and chromatin. Consistent with this expectation, spatial cross-correlation analysis[32] demonstrates that regions containing slow TOP2A are enriched near bound TOP2A loci (Supplementary Fig. 3A, B), with a decay length of ~250 nm, indicating that slow molecules are most likely to be found within this distance from bound ones. Following 1,6-hexanediol treatment, the cross-correlation between slow-diffusing and bound TOP2A molecules diminished by about two-fold relative to the control, indicative of decreased proximity (Supplementary Fig. 3C). We should also be able to visualize the TOP2A equilibrium shift upon ICRF-193 treatment. To test this, we used STED to observe TOP2A association with transcription condensates labeled with MED4 and TOP2A association with open chromatin labeled with H3.3, which we predict to contain TOP2A molecules belonging to the slow and bound fractions, respectively[40] (Fig. 2H). TOP2A was enriched in both compartments, indicating overlap with transcriptional condensates and transcribed chromatin (Fig. 2I). Notably, ICRF-193 treatment shifted TOP2A from the MED4 compartment to the H3.3 compartment (Fig. 2I, J), consistent with its movement from the slow fraction to the bound fraction (Fig. 2C). This shift was not observed with actinomycin D treatment, ruling out the possibility that it was driven solely by increased nucleoplasmic translocation (Fig. 2J, Supplementary Fig. 3D). Altogether these results indirectly support an association between TOP2A and Mediator-containing condensates and suggest that these structures may contribute to the behavior of the TOP2A slow fraction.

Thus, nucleoplasmic TOP2A can be grouped into three states based on diffusion rates: a bound fraction interacting or catalytically engaged with chromatin, a slow fraction enriched in transcription condensates, and a fast fraction diffusing through the nucleoplasm. The equilibrium maintained between these states and the nucleolus enables TOP2A to rapidly shift to regions where topological regulation is needed.

## MYC modulates TOP2A diffusion in cell

We have previously shown that the oncogenic transcription factor c-MYC (hereafter referred to as MYC) is able to join TOP1 and TOP2A to form the topoisome. Within this complex, MYC stimulates the catalytic activity of both topoisomerases, thereby coupling oncogenic transcription rates to relief from DNA entanglements[5]. If the formation of a topoisome prevents the binding of other proteins, thereby limiting the effective size of the complex, as shown in the MYC-driven topoisome study, then we would expect to observe differences in TOP2A diffusion with or without MYC, since protein diffusion is inversely correlated with the complex radius[41].

We generated HCT116 cells expressing MYC tagged with an auxin-inducible degron (AID)[42], enabling 95% knockdown of MYC protein within 90 min of auxin treatment (Fig. 3A). We then tagged endogenous TOP2A via Halo-tag and performed SMT to track TOP2A diffusion upon auxin-dependent MYC degradation. We observed that the proportion of TOP2A found in the fast, slow, and bound fractions was not significantly different between the CTRL and Auxin conditions (Fig. 3B). However, TOP2A diffusion speed was reduced after MYC

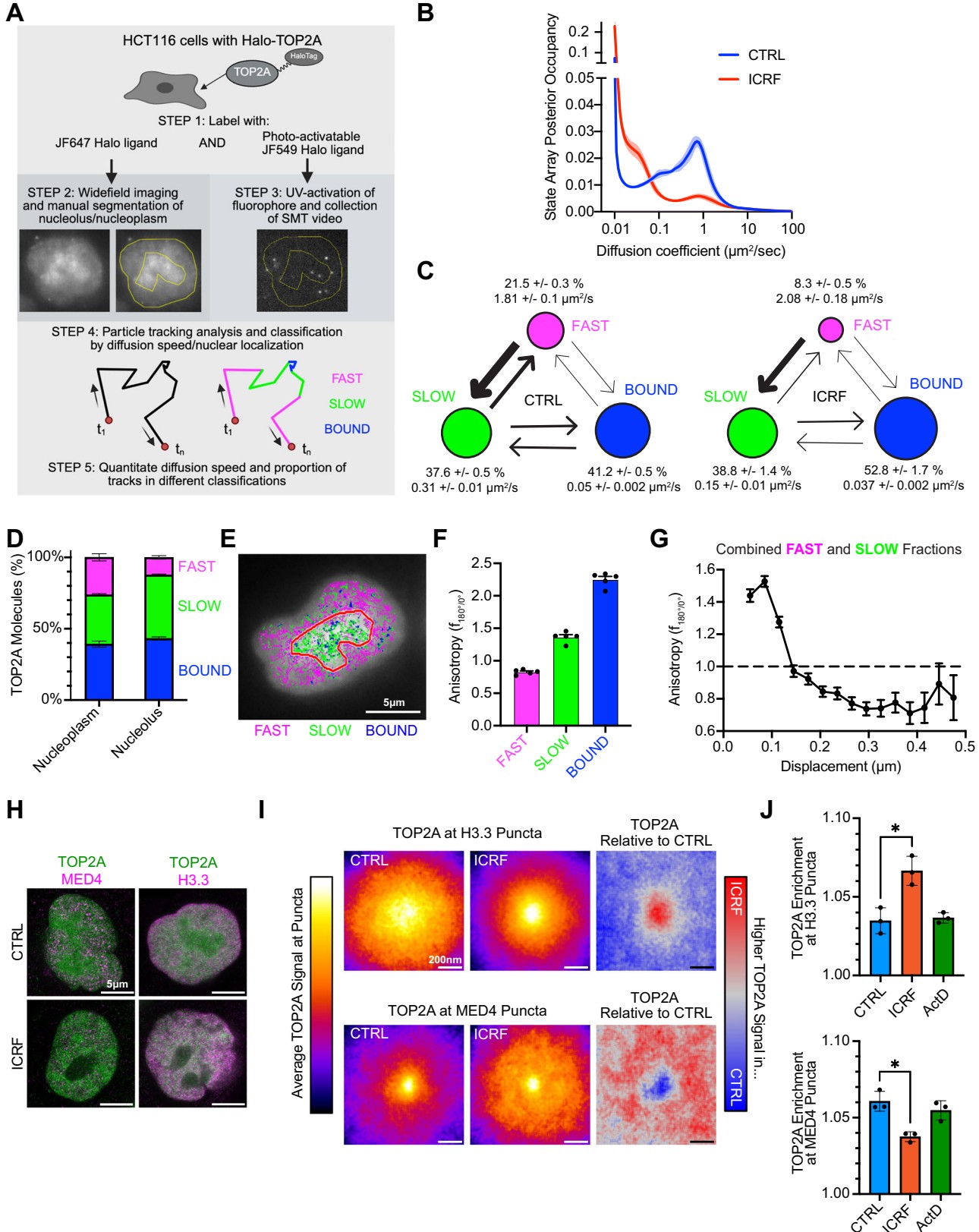

degradation in both the fast and slow fractions of the nucleoplasm (Fig. 3C), while we found no difference in the nucleolus (Fig. 3D) where MYC is largely excluded (Supplementary Fig. 4A). This reduction in TOP2A diffusion was not observed upon global transcription inhibition by triptolide treatment[43], indicating that the diffusion change observed upon MYC depletion was not linked to secondary transcriptional effects (Supplementary Fig. 4B). Indeed, consistent with its ability to reduce transcription-induced supercoiling[44], triptolide increased slow and bound TOP2A diffusion coefficients, potentially by decreasing its DNA engagement, while the unchanged fast fraction coefficient indicates the size of the freely diffusing TOP2A complex remains unaffected (Supplementary Fig. 4B).

**Fig. 2 | TOP2A exists in various diffusive states within different compartments of the nucleus. A** Scheme of SMT detection and analysis. Created in BioRender (Cameron, D. (2026) https://BioRender.com/y3qrwwb). **B** Distribution of TOP2A diffusion coefficients from SMT in Halo-TOP2A HCT116 cells ± 5 µM ICRF-193 for 5 to 45 min, as obtained by SASPT analysis[83]. Average of four experiments, the shaded areas represent SEM. **C** Percentage of TOP2A in Fast, Slow, and Bound fractions and their relative diffusion coefficient in control conditions and upon 5 µM ICRF-193 for 15 min. Width of the arrows is proportional to the frequency of detected transitions between states. Average of four independent experiments, 12 image fields per condition per experiment. **D** Percentage of Fast, Slow and Bound fractions within nucleoplasm and nucleolus. Data are presented as mean values ± SEM from five independent experiments. **E** Representative image of Halo-TOP2A HCT116 cell stained for TOP2A (white) with nucleolus circled in red, and Fast, Slow, and Bound TOP2A tracks highlighted in pink, green, and blue, respectively. Nucleolus is identified by GFP-Nucleolin transfection and detected through multifocus

Structured Illumination Microscopy (mSIM)[32]. **F** Average anisotropy values of Fast, Slow and Bound tracks from Halo-TOP2A HCT116 cells. 5 independent experiments, 115 cells total. Data are presented as mean values ± SEM. **G** Anisotropy values ($f_{180/0}$) vs. mean displacement (µm) of combined Fast and Slow tracks from the nucleoplasm of 115 Halo-TOP2A HCT116 cells from five independent experiments. Data are presented as mean values ± SEM. **H** Representative STED images of HCT116 cells labeled with TOP2A and either MED4 or H3.3 ± 5 µM ICRF-193 treatment for 30 min. **I** Average TOP2A signal at H3.3 (top) and MED4 (bottom) puncta from STED images in control (left) or after 5 µM ICRF-193 treatment for 30 min (center), and the difference between both conditions (right). Signal averaged from three independent experiments. 7177–14486 puncta per condition. **J** Average TOP2A puncta enrichment across independent experiments from data in (**I**) and Supplementary Fig. 3D. *$p < 0.05$ (One-way ANOVA, Dunnett's multiple comparison test, H3.3 $p = 0.046$, MED4 $p = 0.028$).

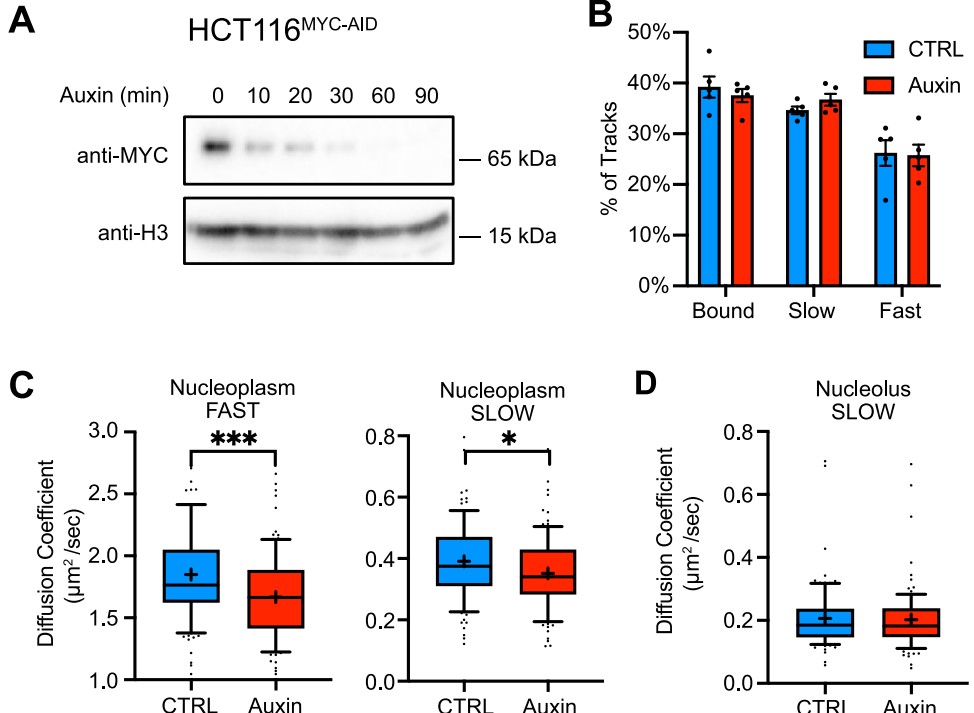

**Fig. 3 | Loss of MYC reduces TOP2A diffusion in cells. A** Representative western blotting demonstrating near complete degradation of MYC in Halo-TOP2A HCT116[MYC-AID] cells within 90 min of treatment with 500 µM auxin. Two independent experiments. **B** Proportion of TOP2A in Fast, Slow, and Bound nucleoplasmic fractions from Halo-TOP2A HCT116[MYC-AID] cells ± auxin treatment. Average of five independent experiments. Data are presented as mean values ±

SEM. **C, D** Diffusion coefficients of Fast nucleoplasmic (**C**, left), Slow nucleoplasmic (**C**, right), and Slow nucleolar (**D**) fractions from individual Halo-TOP2A HCT116[MYC-AID] cells. Average of four independent experiments, 100 cells for each condition. Whiskers extend to the 10th and 90th percentile; the mean value is represented by the + sign. ***$p = 0.0009$; *$p = 0.031$ (Two-tailed unpaired $t$-test).

If MYC promotes formation of comparatively smaller TOP2A complexes by favoring a TOP1-containing topoisome, then removing TOP1 should further increase TOP2A diffusion, as the MYC–TOP2A interaction persists in the absence of TOP1 (Supplementary Fig. 4C–E), yielding an even smaller, more mobile complex. To test this, we depleted free TOP1 either with acute treatments of camptothecin (CPT) to trap and stabilize TOP1 cleavage complex on the DNA[45], or by auxin-mediated degradation in HCT116 cells[5,28] (Supplementary Fig. 4F). In both conditions, TOP1 loss increased TOP2A diffusion, particularly in the slow and bound fractions (Supplementary Fig. 4G, H), potentially indicating that the topoisome forms predominantly within the transcriptional condensates and on the chromatin. Thus, consistent with diffusion scaling inversely with complex size, these results reveal that MYC limits complex size to promote TOP2A diffusion. We next asked whether this phenotype is recapitulated both in vitro and in cells.

## MYC limits TOP2A droplet size

In vitro, we took advantage of recent findings that TOP2A forms phase-separated droplets[8]. The droplets are largely dependent on the presence of the TOP2A-CTD, which is intrinsically disordered. We were able to observe droplet formation using 2 µM recombinant human TOP2A in an isotonic salt solution that was enhanced both by co-incubation with DNA and reduction in salt concentration (Fig. 4A, B, Supplementary Fig. 5A, B), in line with previous reports[8]. These droplets were also able to fuse (Supplementary Fig. 5C). Furthermore, we observed that while supercoiled plasmid DNA and catenated kDNA– both known substrates of TOP2A–as well as RNA could increase droplet formation, short 20 bps DNA oligonucleotides did not (Fig. 4A, B), confirming previous data that DNA was required to be at least 50–100 bps to favor droplets formation[8].

The dependence on DNA length to stimulate droplet formation implies the existence of a percolated network structure within the

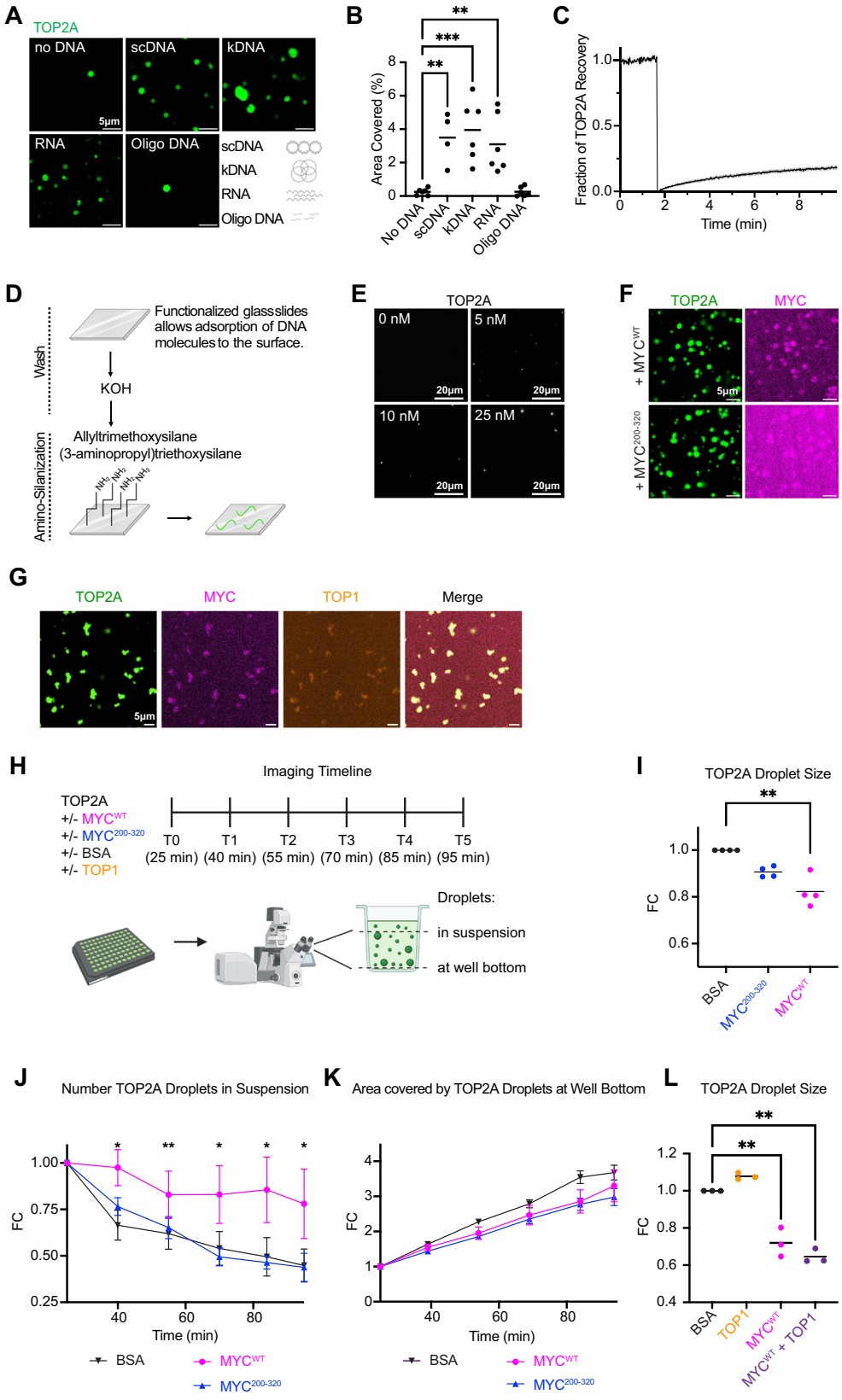

droplets, where physical crosslinks form between molecules, such as through ionic interactions[46]. This contrasts with classical liquid-liquid phase separation (LLPS), where a density transition occurs at the saturation concentration[47]. If these droplets form through LLPS, we would expect to observe free diffusion between the dilute and droplet phases and a strict threshold for determining the concentration where

phase separation occurs. However, we observed limited diffusion between phases as assessed by fluorescence recovery after photobleaching (FRAP) (Fig. 4C), in contrast to near complete fluorescence recovery as seen with liquid-liquid phase-separated droplets formed by the protein FUS[48]. To image TOP2A condensates at protein concentrations below previously reported saturation concentrations, we

**Fig. 4 | MYC affects the biophysical properties of TOP2A condensates in vitro.**
**A** Representative images of 2 μM SNAP-TOP2A condensates in the absence or presence of 10 ng/μl negatively supercoiled plasmid DNA (scDNA), catenated kinetoplast DNA (kDNA), RNA, or 20 bp oligonucleotides (oligo DNA). Scale bar = 5 μm. Created in BioRender (Kuzin, V. (2026) https://BioRender.com/8cutpvz). **B** Quantitation of percentage of area covered by TOP2A droplets in conditions from (**A**), each point representing the per-experiment means (scDNA $n = 4$, others $n = 6$). ***$p < 0.001$; **$p < 0.01$ (no DNA vs. scDNA $p = 0.003$, no DNA vs. kDNA $p = 0.0002$, no DNA vs. RNA $p = 0.004$, no DNA vs. oligo DNA $p > 0.9999$) (ordinary one-way ANOVA). **C** FRAP experiment of 2 μM SNAP-TOP2A condensates in the presence of 10 ng/μl scDNA. Average and ± SEM of 15 bleaches from five independent experiments. **D** Scheme of droplet deposition onto silanized slides used in (**E**) and Supplementary Fig. 5D, K. Created in BioRender (Kuzin, V. (2026) https://BioRender.com/h0774pe). **E** Visualization of SNAP-TOP2A condensates at indicated concentrations after addition of scDNA deposited onto functionalized glass coverslips. Contrast differs between images to demonstrate condensate formation at low concentrations. Scale bar = 20 μm. **F** Representative images of co-incubation of 500 nM SNAP-TOP2A with 3.5 μM SNAP-MYC[WT] or SNAP-MYC[200-320]. Scale bar = 5 μm. **G** Representative images of co-incubation of 500 nM SNAP-TOP2A, 3.5 μM

SNAP-MYC[WT], and 500 nM SNAP-TOP1 in the presence of 10 ng/μM scDNA. Scale bar = 5 μm. **H** Schematic of imaging droplets over time. Created in BioRender (Kuzin, V. (2026) https://BioRender.com/e90w1di). **I** Normalized quantitation of 500 nM SNAP-TOP2A droplet size ± 3.5 μM BSA, SNAP-MYC[WT], or SNAP-MYC[200-320] in the presence of 10 ng/μM scDNA. The plot shows the average of four independent experiments, with each point representing per experiment means. **$p < 0.0074$ (Ordinary one-way ANOVA, Tukey correction). **J, K** Quantitation of SNAP-TOP2A droplets from 2 μM SNAP-TOP2A ± 3.5 μM BSA, SNAP-MYC[WT] or SNAP-MYC[200-320] in suspension (**J**) or at well bottom (**K**) over the course of 95 min, normalized to the 25 min time point. **$p < 0.01$; *$p < 0.05$ (j, 25 min $p = 0.025$, 40 min $p = 0.006$, 55 min $p = 0.026$, 70 min $p = 0.012$, 95 min $p = 0.033$) (multiple paired $t$-tests, no correction). The plot shows the average of six independent experiments and SEM. **L** Normalized quantitation of SNAP-TOP2A droplet size in the presence of 3.5 μM SNAP-MYC[WT], 500 nM SNAP-TOP1, both SNAP-MYC[WT] and SNAP-TOP1, or 3.5 μM BSA. The plot shows the average of three independent experiments, with each point representing per experiment means. **$p < 0.01$ (BSA vs TOP1 $p = 0.994$, BSA vs MYC[WT] $p = 0.018$, BSA vs MYC[WT] + TOP1 $p = 0.007$) (Ordinary one-way ANOVA, Tukey correction).

adapted a protocol where glass cover slips are functionalized to generate a surface with net positive charge that attracts the negatively charged DNA, thus keeping the condensates stationary[49] (Fig. 4D). Using this technique, we could observe droplet formation in the presence of plasmid DNA at protein concentrations as low as 5 nM (Fig. 4E). The variance in puncta size indicated that these droplets are made up of multiple TOP2A molecules, excluding the possibility that we observed individual TOP2A molecules. Additionally, these TOP2A condensates were also observed independently of DNA, ruling out the possibility that the TOP2A molecules were merely connected by binding to the same DNA (Supplementary Fig. 5D, Fig. 4A). These data suggest that TOP2A droplets form by percolation and that TOP2A condensates can still form at sub-saturating concentrations of proteins.

Since MYC can also bind TOP2A independently of TOP1[5] (Supplementary Fig. 4C–E), we tested whether MYC can affect TOP2A condensate formation in vitro. As a negative control, we used BSA as well as a MYC mutant containing only residues 200–320 (MYC[200-320]) that does not interact with TOP2A in immunoprecipitation experiments[5]. Incubation of fluorescently labeled recombinant TOP2A and MYC[WT] with plasmid DNA, showed that MYC co-localizes with TOP2A droplets (Fig. 4F) albeit in a non-stoichiometric manner (Supplementary Fig. 5E), while MYC[200-320] showed only weak co-localization with TOP2A droplets (Fig. 4F, Supplementary Fig. 5F). While others have reported that MYC can phase-separate in the presence of crowding agents such as PEG[10], we did not observe MYC-only droplets under the conditions used here (Supplementary Fig. 5G). We found that fluorescently labeled recombinant TOP1 could colocalize into TOP2A condensates together with MYC (Fig. 4G), while TOP1 showed only a limited capacity to form phase-separated droplets in the presence of DNA, as compared to TOP2A (Supplementary Fig. 5H).

To understand how MYC affects the properties of TOP2A droplets, we devised an imaging strategy that allowed us to observe droplets both in solution and at the well bottom (Fig. 4H). This allowed the visualization of droplets independently of interaction with the plastic surface. We combined 500 nM of fluorescently labeled TOP2A and supercoiled plasmid DNA with 3.5 μM MYC[WT], or MYC[200-320], or BSA in an isotonic buffer and measured the resulting TOP2A droplet properties. MYC[WT] induced a decrease in TOP2A droplet size (Fig. 4I). Similar results were seen at 2 μM (Supplementary Fig. 5I) and 25 nM (Supplementary Fig. 5J) of TOP2A, roughly covering the range of TOP2A nuclear concentrations reported in quantitative mass spectrometry assays[50,51]. In line with the above observation, the TOP2A droplets formed in the presence of MYC[WT] exhibited a markedly slower

rate of sedimentation over a time course of 95 min (Fig. 4J, K) as demonstrated by comparing the relative number of droplets that remain in suspension vs. those at the well bottom. This is in contrast to co-incubation of TOP2A with MYC[200-320], where sedimentation rates were consistent with the BSA control, suggesting that co-localization of MYC[WT] within TOP2A condensates also reduces their droplet density.

We noted that MYC[200-320] elicited a limited decrease in TOP2A droplet size (Fig. 4I, Supplementary Fig. 5I), indicating that it was able to modulate TOP2A droplet formation to some degree, while lacking the region required for direct interaction with TOP2A. However, these experiments were done with much higher concentrations of MYC as compared to pull-down conditions (100 nM) where we demonstrated that MYC[200-320] did not bind TOP2A during immunoprecipitation. We propose this could be due to the presence of MYC[200-320], albeit lower than MYC[WT], in the TOP2A droplets (Fig. 4F). Thus, reducing protein concentration should abrogate this effect. Using the glass coverslip system described previously (Fig. 4D), we demonstrated that at lower protein concentration (25 nM TOP2A and 100 nM MYC), MYC[WT] reduced the size of TOP2A droplets, whereas this effect was negligible with MYC[200-320]. This suggests that the effect of the mutant required a much higher concentration than MYC[WT] (Supplementary Fig. 5J). Note that, despite MYC's ability to interfere with TOP2A droplet formation, MYC-TOP2A interaction does not directly depend on TOP2A's CTD, as shown by the immunoprecipitation of MYC with the TOP2A[ΔCTD] mutant that lacks this domain (Supplementary Fig. 5K, L).

We also examined whether the addition of TOP1 influences droplet behavior. Notably, MYC[WT] also induced a decrease in TOP2A droplet size and an increase in the number of droplets in suspension in the presence of 500 nM TOP1. This indicates that the ability of MYC to reduce TOP2A condensate size and sedimentation is independent of TOP1 and likely affects the entire topoisome. Importantly, droplets containing only TOP2A and TOP1 did not exhibit this reduction, confirming that the effect is MYC-dependent (Fig. 4L, Supplementary Fig. 5M). Together, these data suggest that MYC reduces the size of TOP2A droplets both in the presence and absence of TOP1, potentially by weakening the interactions of TOP2A with other TOP2A molecules.

## MYC limits TOP2A interactions in cells
To test whether MYC regulates the size of TOP2A-containing complexes in cells, we used K562[MYC-AID] cells[42], previously used to study topoisomerases upon MYC acute depletion[5], to rapidly degrade MYC and test whether it affects the ability of TOP2A to establish interaction with other molecules. MYC was almost completely degraded upon 90-

min treatment with auxin, while TOP2A protein levels were unaffected (Supplementary Fig. 6A). Glycerol gradient centrifugations of benzonase-digested protein lysates revealed that acute MYC-depletion caused a shift of TOP2A towards higher molecular-weight complexes (Fig. 5A, Supplementary Fig. 6B-E). Altogether, these findings suggest a mechanism of MYC target regulation, where the presence of MYC is required to accelerate diffusion of TOP2A by limiting the effective size of TOP2A-forming complexes, both outside (fast fraction) and within (slow fraction) transcription condensates.

## MYC promotes substrate detection for TOP2A

Intuitively, increasing enzyme diffusion rates should enhance substrate detection and, therefore, enzyme activity. We directly visualized this using quadruple trap optical tweezers (Q-TRAP) coupled with fluorescent imaging to monitor TOP2A binding at a DNA crossover formed by intertwining two linearized lambda DNA molecules (Fig. 5B). Using this system, we are also able to visualize the product of TOP2A activity by observing the decatenation of the DNA strands. Under physiological conditions including $Mg^{2+}$ and ATP, TOP2A-mediated decatenation occurs extremely quickly (within the ~4 s it takes to image another frame) such that we observe decatenation in more than 72% of instances immediately upon TOP2A binding or without even detecting the bound intermediate (Fig. 5C). The rapid nature of the decatenation reaction, combined with signal-to-noise constraints that arise when further lowering TOP2A concentration, makes it challenging to directly compare decatenation rates with or without MYC in real time using this system. To circumvent this issue, we removed ATP so that decatenation could not occur, thereby prolonging the observation window. We then collected images at 30 s intervals and measured TOP2A droplet binding at the crossover by applying an intensity threshold (Fig. 5D). Under these conditions, we detected more frequent TOP2A binding events when MYC was present (Fig. 5E). Given that increased substrate binding would logically translate into higher catalytic turnover when ATP is available and, considering our prior demonstration that MYC stimulates TOP2A activity in standard assays[5], these data strongly suggest that the MYC-enhanced TOP2A diffusion promotes greater substrate engagement, thereby increasing enzymatic activity.

## MYC increases TOP2A activity in cells

If MYC promotes TOP2A enzymatic activity by favoring substrate detection, decreasing MYC levels should decrease TOP2A-DNA cleavage complexes (TOP2Accs) in cells. To detect TOP2Accs genome-wide, we optimized our previously developed CAD-seq[52] for TOP2A to immunoprecipitate only catalytically engaged TOP2A. HCT116[MYC-AID] cells were treated sequentially with auxin, proteasome inhibitor MG132, and—in the last 6 min—etoposide to trap and stabilize TOP2ccs[45,53] (Fig. 5F). MG132 was included to prevent TOP2cc degradation[53]. In control cells, TOP2Acc accumulated at transcription start sites (TSS) and downstream of transcription end sites (TES) of the top 1500 (~10%) expressed genes (Fig. 5G, Supplementary Fig. 6F) as previously observed[54]. With MYC degradation, TOP2Accs were markedly reduced across all gene locations, particularly towards the TESs, indicating that TOP2A catalytic engagement on the DNA is MYC-dependent. This was not due to a general downregulation of transcription, as 90 min of auxin treatment caused essentially no change in transcript production either globally or among the top 10% most highly expressed genes, as assessed by SLAM-seq, which metabolically labels nascent transcripts genome-wide[42] (Fig. 5H, Supplementary Fig. 6G, H). Altogether, these findings demonstrate that MYC limits the size of TOP2A complexes, increasing diffusion and substrate detection, which affects TOP2A catalytic engagement on the DNA (Fig. 6). Without ruling out the possibility of additional mechanisms, we suggest that these results reveal the primary mechanism by which MYC stimulates TOP2A.

## Discussion

We demonstrate that TOP2A exists in a multipartite equilibrium maintained by distinct nuclear condensates and is acutely sensitive to dysregulation of topoisomerase activity (Figs. 1, 2). There are many potential advantages to this equilibrium. For instance, TOP2A levels must be strictly regulated throughout the cell cycle to prevent aberrant activity. Its expression increases during S-phase, peaks in mitosis, and is subsequently degraded[55,56]. This ensures sufficient TOP2A is available during mitosis where it is required to condense and disentangle chromatin[57–59], while rapid degradation prevents toxicity from excess cellular TOP2A during interphase[60]. Thus, the cell must require a mechanism to sequester excess TOP2A when expression is upregulated during G1/S-phase while also facilitating rapid degradation upon mitotic exit.

We find that TOP2A is sequestered away from nucleoplasmic DNA into the nucleolus through an RNA-dependent mechanism (Fig. 1). Previous work showed that residues 1192-1289 in the TOP2A CTD are required for RNA binding and nucleolar enrichment[22]. This region also confers sensitivity to RNA-mediated activity inhibition[22], suggesting that RNA-rich compartments, such as the nucleolus, may repress TOP2A activity. Additionally, the nucleolus is a hub for degradation of misfolded and excess proteins[61,62], implying it could buffer excess TOP2A by sequestration and facilitate protein degradation after mitosis. Further experiments combining quantitative imaging with acute control of TOP2A expression would be valuable to test this hypothesis. Intriguingly, TOP2B displayed a more homogeneous nuclear distribution compared with TOP2A. We ascribe such a difference to their CTDs, which show lower sequence conservation (~46%) as compared to the rest of the proteins (~77%)[63]. The molecular differences between the CTDs, including their distinct distributions of charged amino acids and isoelectric points[8], may underlie the more pronounced nucleolar-to-nucleoplasmic redistribution observed for TOP2A relative to TOP2B.

Diffusion in the cell is far slower than would be expected for macromolecules of similar size in a dilute solution. Molecular dynamics simulations of crowded protein environments analogous to the nucleus indicate that reduced diffusion arises from proteins forming transient clusters or complexes with other proteins[64]. These simulations demonstrate that diffusion of an individual macromolecule decreases proportionally with the number of contacts it makes with other molecules in its environment, causing a broad range of diffusion values for the same protein[65].

We similarly detect a wide distribution of TOP2A diffusion coefficients (Fig. 2B). Although we group these rates into three classes (Fig. 2C) for comparison, each class likely contains multiple TOP2A sub-states. In the bound fraction, for instance, we cannot distinguish between catalytically engaged TOP2A vs. TOP2A merely interacting with chromatin. Since the presence of MYC increases catalytic engagement of TOP2A on genomic DNA (Fig. 5G), one might expect an increase in the TOP2A-bound fraction. However, since MYC limits non-specific TOP2A self-interaction in vitro (Fig. 4), it may also reduce TOP2A's non-specific chromatin interactions. These counteracting effects on TOP2A-DNA interactions may explain why we could not observe a difference in the proportion of bound TOP2A +/- MYC as measured by SMT (Fig. 3B).

Moreover, while we link the slow fraction of TOP2A to transcription condensates based on anisotropy (Fig. 2F, G) and co-localization with Mediator (Fig. 2H–J), TOP2A diffusion likely depends on numerous specific and non-specific binding partners in its microenvironment. Indeed, TOP2A interacts with hundreds of proteins[66], as well as DNA. Advances that improve both the throughput and resolution of SMT[67] could enable further distinction of these sub-states and help define the mechanism of TOP2A diffusion in greater detail.

We also show that TOP2A diffuses faster in the nucleoplasm in the presence of MYC (Fig. 3). This finding suggests that the average TOP2A

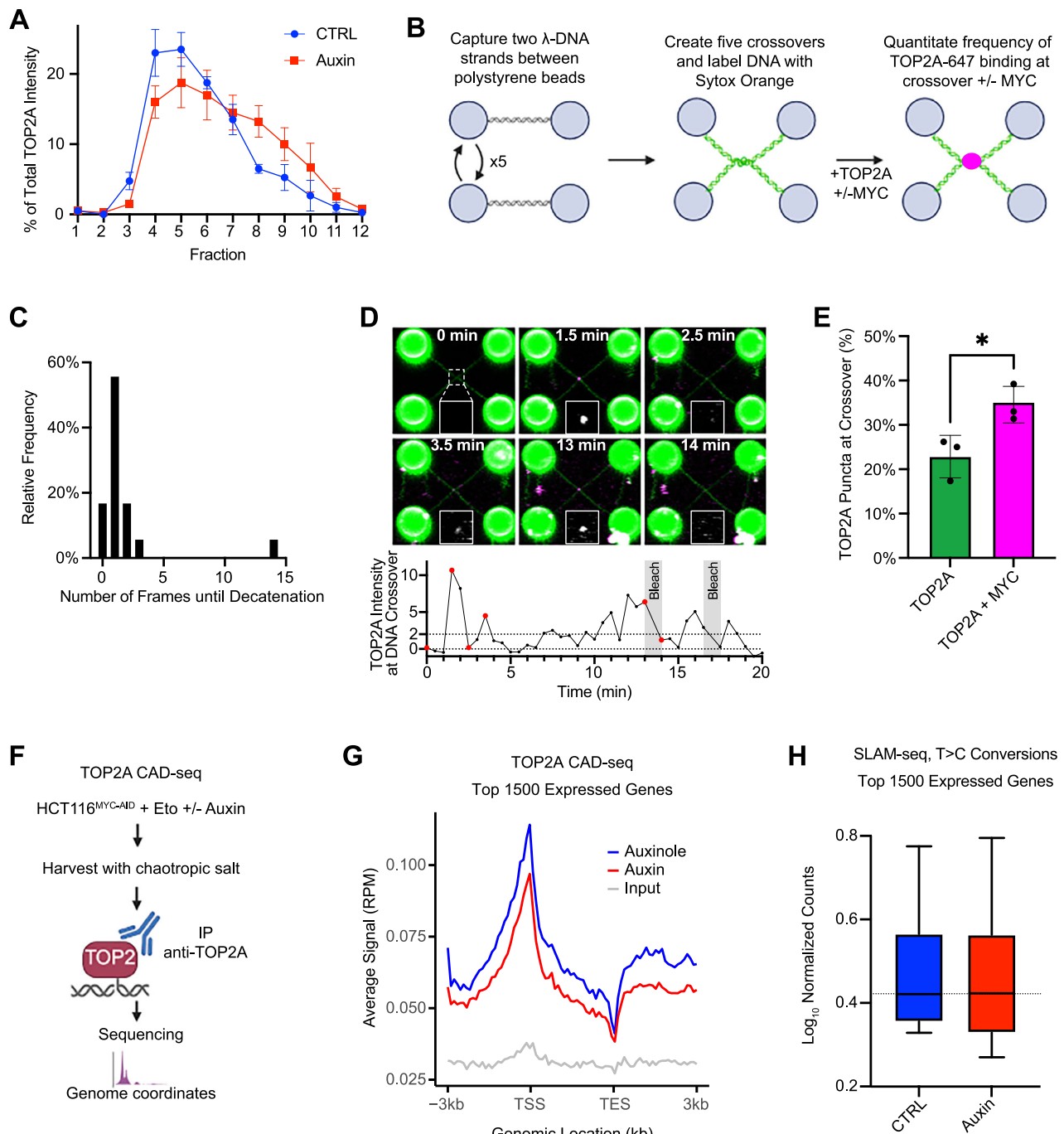

**Fig. 5 | MYC limits TOP2A complex size favoring substrate detection and TOP2A activity. A** Quantitation of glycerol gradient experiments from Supplementary Fig. 6B–E. Band intensity is normalized as a fraction of the total intensity across fractions, to control for variation of total protein amounts. The plot shows the average of four independent experiments, and SEM normalized to total TOP2A intensity across fractions. **B** Schematic of Q-TRAP experiment. Created in BioRender (Kuzin, V. (2026) https://BioRender.com/tj6plh5). **C** Number of frames (imaged every 4 s) that the DNA crossover persists after detection of TOP2A at crossover from Q-TRAP experiment in the presence of ATP (value = 0 when no TOP2A was visualized before crossover resolution). Two independent experiments, 8–10 crossovers measured per experiment. **D** Example of TOP2A intensity measurement for DNA crossover after incubation in the absence of ATP. TOP2A intensity at crossover is normalized to the background signal and quantitated every 30 s. TOP2A that persisted at crossover for more than 90 s was bleached to enable visualization of new TOP2A molecules binding to crossover. Snapshots (top) are from time points highlighted in red on the graph (bottom). BSA is included in all conditions to ensure

protein concentration remains consistent. **E** Proportion of images where TOP2A is bound at the DNA crossover in the absence of ATP ± MYC. Three independent experiments, 3–6 crossover measured for each condition per experiment. Data are presented as mean values ± SD. *$p = 0.022$ (Two-tailed paired $t$-test). **F** Scheme of TOP2A CAD-seq. Created in BioRender (Baranello, L. (2026) https://BioRender.com/nat7rze). **G** Metagene plot of TOP2Acc enrichment across the top 1500 ( -10%) expressed genes based on the SLAM-seq in HCT116[MYC-AID] cells ± 90 min auxin. Input DNA used as a negative control. Average of three independent replicas. RPM refers to reads per million. TSS refers to the transcription start site. TES refers to the transcription end site. See individual heatmaps in Supplementary Fig. 6F. **H** Spike-in normalized SLAM-seq read counts containing T > C conversions in the top 1500 expressed genes as defined from the SLAM-seq experiments in HCT116MYC-AID cells ± 90 min auxin. Average of three independent replicas. Log[10] normalized counts refer to human T > C counts normalized to ERCC filtered reads and division by gene length (kb). Data were then log[10]-transformed after addition of a pseudo-count (+ 1). Whiskers represent 10th and 90th percentiles. Outliers are not shown.

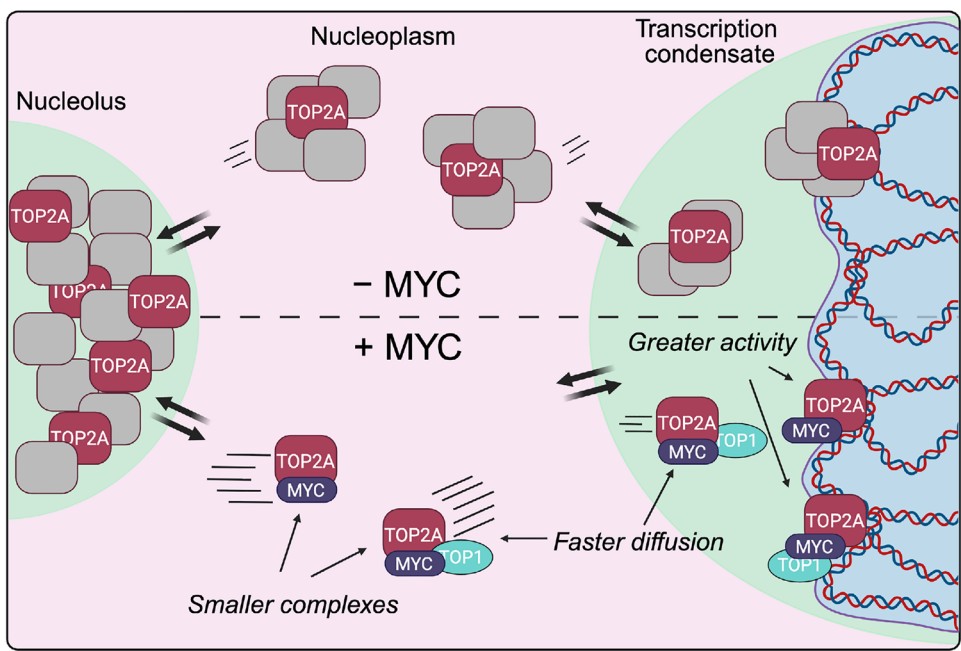

**Fig. 6 | Model: MYC acts as an accelerant of TOP2A in the nucleoplasm.** TOP2A exists in an equilibrium between the nucleolus, nucleoplasm, and transcription condensates, which is highly responsive to changes in supercoiling. The diffusion of TOP2A can be categorized into Fast (pink), Slow (green) or Bound (blue), and is dependent on the cellular compartment. In the presence of MYC, TOP2A forms protein complexes, including the topoisome with TOP1, that are, on average, smaller than in the absence of MYC, suggesting that MYC limits TOP2A interaction with other proteins (gray boxes). This causes TOP2A to diffuse faster and increases its chromatin binding, thus stimulating its activity. Created in BioRender (Cameron, D. (2026) https://BioRender.com/a1vz4uz).

complex size is smaller when interacting with MYC. Upon down-regulation of MYC by auxin treatment, the diffusion coefficients of fast and slow TOP2A molecules are approximately 0.85x to 0.9x smaller than in the presence of MYC. According to the Stokes-Einstein relationship, in the first approximation, diffusion coefficients would scale as the cubic root of the diffusing complex, thus corresponding to roughly 1.4x to 1.6x larger complexes upon MYC degradation. We confirm this shift by glycerol gradient fractionation (Fig. 5A) and by measuring TOP2A droplet size as a proxy for TOP2A self-interaction (Fig. 4D–L). Notably, the reduction in droplet size is observed also in the presence of TOP1 (Fig. 4L). These results align with our prior characterization of a size-limited topoisome comprising TOP2A, TOP1, and MYC, which stimulates topoisomerase activity in vitro and in cells[5]. Recently, an alternative topoisome where topoisomerases are nucleated and stimulated by p53 has also been identified[68]. Given that TOP2A interacts with many proteins, there may be multiple "flavors" of topoisome, each conferring distinct impacts on topoisomerase activity. We hypothesize that changes in TOP2A diffusion, as shown here with MYC, partially explain these distinct activity profiles. Future studies should integrate enzymatic assays with measurements of complex size and diffusion to test the generality of this mechanism.

Our data indicate that MYC functions as an "accelerant", favoring TOP2A diffusion and thereby increasing substrate detection. For a distributive enzyme such as TOP2A, increasing the likelihood of finding its preferred substrate increases activity, which we detect at sites of high transcriptional output (Fig. 5G). Acceleration of transcription factor exchange is also associated with increased transcriptional output, highlighting the benefit of increased diffusion rates[69]. Notably, cells with greater metastatic potential have lower cytoplasmic viscosity and higher diffusivity than those with lesser metastatic potential, implying that increased diffusion may be linked to malignancy[70,71]. MYC's established role as a transcriptional amplifier is grounded in its ability to recruit transcription factors to their targets, but how exactly it achieves this remains unclear. Indeed, MYC has many interacting partners[72], including other enzymes involved in RNA processing, DNA replication, and chromatin modification—all processes understood to be stimulated by MYC independently of its DNA binding functions[72–74]. Interestingly, DNA-binding-deficient MYC induces proliferation in some systems[73], but not others[75], raising the question of whether its transcriptional function can be separated from its accelerant role. Future studies are needed to determine whether other proteins also enhance diffusion in a manner similar to MYC and to explore how cells exploit diffusion control to regulate enzymatic function.

## Methods
### Cell lines
The human colon carcinoma cell line HCT116 (ATCC, CCL-247) and variants thereof are used throughout this manuscript. Halo-TOP2A HCT116[MYC-AID] cells were generated by knocking in an N-terminal Halo-tag into both TOP2A alleles of HCT116 cells expressing MYC tagged with an Auxin-Inducible Degron (AID) system[42] (gift of Dr. Zuber, (IMP, Austria)). These cells were subsequently transduced with a BFP-labeled OsTir1 delivery vector[42] to enable MYC degradation. Halo-TOP2A HCT116[TOP1-AID] cells were generated by knocking in an N-terminal Halo-tag into both TOP2A alleles of HCT116 CMV-OsTIR1 mAID-TOP1 cells[28] (here referred to as HCT116[TOP1-AID]). All HCT116 cells were cultured in high glucose (4.5 g/L) DMEM (Thermo Fisher, 61965059) with 10% Fetal Bovine Serum (Thermo Fisher, 10270106) in a 37 °C incubator with 5% $CO_2$. Halo-TOP2A HCT116[TOP1-AID] cells[28] were additionally cultured with 1 µg/ml puromycin (Thermo Fisher, A11138-03) and 125 µg/ml hygromycin B (Thermo Fisher, 10687010). K562[MYC-AID] cells (gift from Dr. J. Zuber (IMP, Austria))[42] were cultured in RPMI 1640 medium (Thermo Fisher, 21875034) supplemented with 10% Fetal Bovine Serum (Thermo Fisher, 10270106) and 2 mM L-Glutamine (Thermo Fisher, 25030081), 1 mM sodium pyruvate (Thermo Fisher, P5770598) in a 37 °C incubator with 5% $CO_2$. AID-mediated degradation was induced by treatment with 500 µM auxin (3-indoleacetic acid, Merck, I3750).

## Recombinant protein expression

Recombinant human TOP2A labeled at the N-terminal with the SNAP-tag sequence[76] was expressed from the 12-URA-B plasmid in yeast cells and purified as previously described[77]. Briefly, the plasmid was transformed into URA-deficient yeast and grown in uracil-deficient media to select for transformed cells. Yeast cultures were grown in YPLG media (1% yeast extract, 2% peptone, 2% sodium DL-lactate, 1.5% glycerol) overnight, and TOP2A expression was induced by the addition of 2% galactose for 6 h. The protein lysate was extracted by cryo-milling, filtered, and TOP2A was enriched using sequential HisTrap nickel and HiTrap CP cation exchange columns before cleavage with His-tagged TEV protease overnight. The TEV and cleaved His-tag were removed by repassage through a HisTrap nickel column, and the SNAP-TOP2A was purified using a Superdex 200 16/60 column, concentrated by filter centrifugation, and stored at −80 °C. SNAP-tagged TOP2 activity was tested by relaxation/decatenation assay, confirming that the tag does not alter enzyme catalysis.

For expression of TOP2A$^{\Delta CTD}$, the previously published[78] expression vector coding for the wild-type human TOP2A was mutated by site-directed mutagenesis to introduce a STOP codon at residue 1217 using the QuikChange XL Site-Directed Mutagenesis kit (Agilent), generating the hTopo IIA Δ1217plasmid. TOP2A$^{\Delta CTD}$ was overexpressed in Baby Hamster Kidney cells (BHK21). TOP2A$^{\Delta CTD}$ was first purified on a StrepTrap HP column (Cytiva, 23151816). The protein was washed with 25 mM Hepes, 100 mM NaCl, 100 mM KCl, 1 mM MgCl$_2$, 10% v/v glycerol, 2 mM DTT, pH 8 and eluted with the same buffer supplemented with 3 mM Desthiobiotin (Merck, 533-48-2). The Twin-strep tag was removed by the addition of P3C (PreScission protease, Merck, 27-0843-01) at a 1:50 ratio (w/w) and incubated overnight at 4 °C. The cleaved protein was then loaded on a HiTrap Heparin HP column (Cytiva, 17-0407-01). Elution was performed by a single step using 25 mM Hepes, 370 mM NaCl, 370 mM KCl, 1 mM MgCl$_2$, 10% v/v glycerol, 2 mM DTT, pH 8.

Recombinant human MYC$^{WT}$ and MYC$^{200-320}$ with N-terminal SNAP-tag were expressed in Rosetta-gami competent cells in LB media. Expression of MYC was induced with 1 mM IPTG for 4 h, and cells were lysed by probe sonication. Protein was resuspended in buffer (500 mM NaCl, 20 mM Tris pH 8, 35 mM imidazole, 1% Triton X-100) with increasing urea (0 M then 1.6 M) before being solubilized in buffer (NaCl, 20 mM Tris pH 8, 35 mM imidazole, 1% Triton X-100, 7 M urea). Lysate was centrifuged, filtered before passing through a HisTrap column (Merck, GE17-5319-01), concentrated and dialyzed to remove urea before being stored at −80 °C.

Full-length human TOP1 (gift from Yves Pommier, NIH/USA) was cloned into the pFastBac1 vector with either N-terminal His6-SNAP-tag or C-terminal SNAP-tag-His6 coding sequence. Assembled plasmids were transformed into DH10Bac competent cells, and the resulting recombinant bacmids were identified using blue-white colony screening. Isolated bacmids were transfected into Spodoptera frugiperda (Sf9) cells using FuGENE HD transfection reagent (Promega, E2311). Baculovirus was gradually amplified to P3 to achieve a higher titer before initiating expression. To express both SNAP-tagged TOP1 constructs, ~1.5 × 106 cells/mL were incubated with 1:50 P3 virus at 27 °C while maintaining 115 rpm orbital shaking for 3 days. Cells were harvested and lysed by sonication in a lysis buffer (20 mM HEPES, pH 7.3, 300 mM NaCl, 5 mM MgCl2, 5% Glycerol, 0.5 mM TCEP, 10 mM imidazole) supplemented with cOmplete™, EDTA-free Protease Inhibitor Cocktail (Roche, 04693132001). Soluble fraction was obtained by centrifugation at 17,000 x $g$ for 40 min and subsequently incubated with Ni-NTA Agarose resin (QIAGEN, 30210) for 30 min, while rotating in a cold room. Loosely bound protein was washed off with 50 mM imidazole, and the TOP1 proteins were eluted with 300 mM imidazole prior to size-exclusion chromatography. Eluted fractions were loaded onto the ÄKTA pure™ system (Cytiva) coupled with S200 10/300 GL pre-equilibrated in 20 mM HEPES, pH 7.3, 300 mM NaCl, 5 mM MgCl2, 5% Glycerol, 0.5 mM TCEP. Fractions containing TOP1 were concentrated to 1–2 mg/mL using Vivaspin® Turbo 4 MWCO 50,000 (Sartorius, VS04T31), and flash-frozen before further use. SNAP-tagged TOP1 activity was tested by relaxation assay, confirming that the tag does not alter enzyme catalysis.

## Droplet assays

N-terminal SNAP-tagged TOP2A was labeled by incubating with 5 µM SNAP-surface Alexa Fluor 488 (NEB, S9129S) at 37 °C for 30 min. Excess fluorophore was removed through dialysis for 24 h into 500 mM KCl, 20 mM Tris, pH 8, 10% glycerol, 0.5 mM TCEP buffer. C-terminal SNAP-tagged TOP1 was labeled by labeled in 5 µM SNAP-surface Alexa Fluor 546 (NEB, S9132S) at 37 °C for 30 min. Excess fluorophore was removed through dialysis for 24 h into 100 mM KCl, 20 mM Tris, pH 8, 1 mM EDTA, 20% glycerol, 1 mM DTT buffer. N-terminal SNAP-tagged MYC$^{WT}$ and MYC$^{200-320}$ were labeled in 5 µM SNAP-surface Alexa Fluor 647 (NEB, S9136S) at 37 °C for 30 min before being dialyzed into 20 mM Tris pH 8, 100 mM KCl, 1 mM DTT, 1 mM EDTA, 20% glycerol. BSA samples were treated in the same manner as the MYC samples.

Droplet assays done with 500 nM or 2 µM TOP2A and 3.5 µM MYC$^{WT}$, MYC$^{200-320}$, or BSA were mixed with 10 ng/ul pUC19 plasmid DNA or water and allowed to settle for 25 min before imaging over a total of 95 min. Where relevant, 500 nM SNAP-TOP1 was added. Assays done using 500 nM TOP2A were imaged in six locations per well once every 25 min, and images were taken at the well bottom and 6 additional z-stacks with increments of 10 µm. Assays done using 2 µM TOP2A were imaged in one location per well every 5 min, once at the bottom of the well and again focusing 25 µm higher within the well. Imaging was done using a Nikon CrEST X-Light V3 spinning disk microscope. Analysis was done using CellProfiler[79]. Droplet assays were performed in 384-well black non-binding µCLEAR microplates (Greiner, 781906).

## Confocal imaging and immunofluorescence

For all treatments, cells were seeded on 18 mm coverslips (Marienfeld, 630-2200) in 12-well plates to reach 70% confluency by the time of fixation.

For TOP2A localization after ICRF-193 or Actinomycin D treatments, HCT116 cells were treated with 5 µM ICRF-193 (Enzo Life Sciences, BML-GR332) for 15 min, or 5 nM Actinomycin D (Merck, A1410) for 1 h. After treatments, cells were washed once with PBS before being fixed with 3% paraformaldehyde (Thermo Fisher, 28908) for 5 min at room temperature and permeabilized with 100% methanol at −20 °C for 2 min.

For TOP1 degradation and etoposide (Merck, E1383) treatments, HCT116$^{TOP1-AID}$ cells were treated with 500 uM auxin for 3 h to induced TOP1 degradation, or degradation was blocked using 0.1 mM auxinole (MedChemExpress, HY-111444) overnight. Etoposide (1 µM) treatments were performed for 1 h. Cells were fixed and permeabilized as described above.

For CSK treatments, cells were washed with PBS (Thermo Fisher, 14190169). Cells were either incubated with 500 ul CSK buffer (10 mM PIPES, 100 mM NaCl, 3 mM MgCl2, 300 mM sucrose, 0.1% Triton X-100) with or without 300 ng/ml RNase A (Thermo Fisher, 12091021) at room temperature for 5 min or immediately fixed in PFA. All cells were fixed in 3% paraformaldehyde and washed twice in PBS.

After treatments and fixation, cells were blocked in 2% BSA in PBS-T (0.1% Triton X-100 in PBS) for 1 h and then incubated in primary antibodies against: TOP2A diluted 1:200 (Abcam, ab52934), fibrillarin diluted 1:150 (Santa Cruz Biotechnology, sc-166001; or Abcam, ab184817), MYC diluted 1:200 (Abcam, ab32072), anti-γH2AX diluted 1:200 (Merck, 05-636), or TOP2B diluted 1:200 (Santa Cruz Biotechnology, sc-365071) in PBS-T with 2% BSA. Coverslips were then washed three times in PBS-T and incubated in anti-rabbit (Thermo, A32790) or anti-mouse (Thermo, A32744) fluorescently conjugated

secondary antibodies diluted 1:500 and DAPI diluted 1:1000 (Thermo Fisher, D1306) in PBS-T with 2% BSA. Finally, coverslips were washed three times in PBS-T and mounted using ProLong Diamond Antifade Mountant with DAPI (Thermo Fisher, P36966) and sealed with nail polish or mounted using ProLong Diamond Antifade Mountant (Thermo Fisher, P36970) and left to cure overnight.

TOP2A nucleolar to nucleoplasmic intensities were calculated using CellProfiler version 4.2.8. Fibrillarin staining was used to identify nucleoli, and DAPI staining was used to identify the whole nucleus. Nuclei were masked with nucleolar objects to obtain the nucleoplasmic signal. TOP2A (or TOP2B) intensity was measured in the nucleoli and nucleoplasm, and mean nucleolus intensity was divided by mean nucleoplasmic intensity for each cell. Where applicable, the γH2AX signal was instead measured in the nucleoplasm or nucleoli.

Cell cycle staging was enabled by incubation of HCT116 cells seeded to 70% confluency with 20 μM EdU (taken from Invitrogen, C10424) for one hour, before fixation and labeling of TOP2A, fibrillarin, and DAPI as described above. 3D images of the cells were collected, and the z-stacks were averaged in 2D to ensure that the DAPI signal throughout the cell was measured. The DAPI signal was plotted by histogram to determine cutoffs for G1 and G2 peaks. Cells with EdU staining between these cutoffs were considered in S phase. After categorizing cells into cell cycle stages, TOP2A intensity and nucleolar:nucleoplasm ratio were calculated as described previously.

Cells were imaged on Zeiss LSM 700, 710, 880, 980-Airy confocal microscopes using a 63x magnification objective. Graphs and statistical analysis were done on GraphPad Prism version 10.4.1.

### Fluorescence recovery after photobleaching (FRAP)

Recombinant SNAP-tagged TOP2A was labeled with SNAP-surface Alexa Fluor 488, as described above, before being aliquoted and stored at −80 °C. 500 nM TOP2A, 10 ng/ul pUC19 plasmid DNA, and 3.5 μM MYC$^{WT}$ were combined in 384-well black non-binding μCLEAR microplates (Grenier, 781906), and droplets were allowed to form and settle for 10 min before bleaching. Whole droplets were bleached with a 488 nm laser at 100% intensity for 75 iterations. Bleached droplet intensity was measured over 8 min, and droplet intensity was normalized to that of a second, unbleached droplet and plotted with the average pre-bleach signal defined as 1 and the first measurement post-bleach defined as 0. Values measured in instances where unbleached droplets passed in front of either measured droplet were excluded. Intensity measurements were taken using ImageJ. FRAP was performed on a Zeiss LSM 980-Airy confocal microscope using a 63x magnification objective. Graphs and statistical analysis were done on GraphPad Prism version 10.4.1.

### Cover glass functionalization, sample preparation and fluorescent wide-field imaging

DNA-protein complexes were deposited on functionalized microscope coverslips following published protocols[49,80,81]. Coverslips and cover glasses were washed by sonication in a 2% Hellmanex solution, rinsed 3 times with MilliQ (MQ) water, dried with N$_2$-gas, rinsed with acetone, submerged in Allyltrimethoxysilane (ATMS, 95%, Merck), (3-aminopropyl) triethoxysilane (APTES, ≥ 98%, Merck), and Acetone (Merck) at a 1:1:100 ratio for 2 h. Prior to loading a sample, the coverslips were rinsed with MQ water and dried with N$_2$-gas. An 8 μL sample was used per coverslip. A reaction consisting of TOP2A, MYC, and DNA in 50 mM Tris pH 8, 150 mM NaCl, 10 mM MgCl$_2$, 0.5 mM DTT, and 30 μg/mL BSA was set up by equilibrating buffer and protein on ice for 10 min. After adding the DNA, the reaction was incubated at 37 °C for 10 min, and Sytox Orange (SYTOX™ Orange Nucleic Acid Stain, Invitrogen, S11368) was added. The sample was imaged immediately.

Images were taken with an inverted fluorescence microscope (Zeiss AxioObserver.Z1) equipped with a 63x oil immersion objective (NA = 1.46, Zeiss), 1.6×optovar magnification changer, an iXon EMCCD

camera (Andor) and an LDI-7 Laser Diode Illuminator (89 NORTH) was used. The sample was alternately excited with 640 nm, 555 nm, or 470 nm light and filtered with Zeiss Filter set 50, 43, or 44, respectively.

Images were processed with a custom Python script. Briefly, the images were flat-field corrected (FFC), filtered with a Gaussian and a top-hat filter and segmented by joining the result from a local thresholding algorithm and a difference-of-gaussian blob detector. The background was defined as the inverse of the puncta mask. Puncta statistics were extracted from the FFC images.

### Glycerol gradients

K562$^{MYC-AID}$ cells were treated with 0.1 mM auxinole for 24 h, to block MYC degradation, or 500 μM auxin for 90 min. Cells were counted using Countess 3 cell counter (Invitrogen), and 8 million cells were harvested, spun down for 3 min at 300 × g, washed once in PBS (Thermo Fisher, 14190169), and washed once in HWB (20 mM HEPES pH 7.5, 137 mM NaCl). Cells were then lysed in HLB (10 mM HEPES pH 7.5, protease inhibitor tablet (cOmplete, EDTA-free protease inhibitor cocktail, Merck, 4693132001), 0.5 mM AEBSF (4-(2-Aminoethyl) benzenesulfonyl fluoride hydrochloride, Merck, A8456), 10 μM Leupeptin (Leupeptin trifluoroacetate salt, Merck, L2023), 1 μM Pepstatin A (Merck, P5318), and 7 mM NaF) and passed through a 20 G needle 10 times to disrupt the cell membrane before being spun down for 5 min at 800 × g. The resulting pellet was washed again with HLB before being spun again and resuspended in 0.2% sarkosyl, 20 mM HEPES pH 8, 133 mM NaCl, 2 mM MgCl$_2$, 0.4 mM EDTA. Chromatin was sheared through passage with a 25 G needle 7 times, sonication for 5 min, and treated for 20 min with 250 units of benzonase (Merck, E8263) before being centrifuged to remove the remaining insoluble fraction. Samples were loaded into a 10–30% glycerol gradient (equal parts 10 or 30% glycerol, 20 mM HEPES pH 8, 133 mM NaCl, 2 mM MgCl$_2$, 0.4 mM EDTA) created in Polyclear Open Top Ultracentrifuge Tubes (Biocomp Instruments, 151–514 A) using a BioComp gradient master, and spun for 16 h at 151263 × g at 4 °C in a Beckman Colter Optima L-90K ultracentrifuge fitted with a SW41 rotor. Gradients were manually fractionated into 24 fractions and precipitated using 0.18% DOC for 10 min followed by the addition of 9% TCA for 30 min. The fractions were centrifuged at 21130 × g for 10 min, washed twice with 25% acetone, and the protein pellet was resuspended in 1x Bolt LDS (Invitrogen, B007) with 5% β-mercaptoethanol.

### Western blotting

Samples were loaded onto Bolt Bis-Tris Plus Mini Protein Gels (Invitrogen, NW04127BOX) alongside PageRuler Plus Prestained Protein Ladder (Thermo Fisher, 26619), run for 70 min at 160 V, and transferred onto nitrocellulose membrane overnight at 20 V. The membrane was blocked in 3% BSA in TBS-T or 5% milk in TBS-T. Incubation with anti-TOP2A (Abcam, ab52934), anti-MYC (Abcam, ab32072), anti-H3 (Abcam, ab1791), anti-beta actin (Merck, A5441), or anti-TOP1 (Abcam, ab109374) primary antibodies was done in 3% BSA in TBS-T diluted 1:10,000 for 1 h or overnight. Incubation with secondary antibodies (anti-rabbit Jackson ImmunoResearch, 211-032-171; or anti-mouse Jackson ImmunoResearch, 115-035-174) was done in 3% BSA in TBS-T diluted 1:35000 for 1 h. The blot was imaged using SuperSignal West Femto Maximum Sensitivity Substrate ECL (Thermo Scientific, 34095) in a Biorad Chemidoc MP Imaging System. Quantification was done using Image Lab Version 6.1.0 build 7.

### Coimmunoprecipitation of recombinant proteins

Coimmunoprecipitations (coIPs) with recombinant proteins were carried out by mixing 200 ng of full length TOP2A or C-terminally truncated TOP2A (ΔCTD) with 100 ng of MYC protein in PDB buffer (10 mM Tris-HCl pH 7.5, 100 M KCl, 0.5% NP40, 7.5% Glycerol, 200 μM EDTA, 250 μg/ml BSA, 1× protease inhibitor tablet (cOmplete, EDTA-free protease inhibitor cocktail, Merck, 4693132001), 0.5 mM AEBSF

(4-(2-Aminoethyl) benzenesulfonyl fluoride hydrochloride, Merck, A8456), 10 μM Leupeptin (Leupeptin trifluoroacetate salt, Merck, L2023), and 1 μM Pepstatin A (Merck, P5318)) on ice for 30 min. 100 ng of anti-TOP2A/B (Abcam, ab109524), anti-MYC (Abcam, ab32072) or IgG control (Santa Cruz, sc-2025) was added to the mixture, followed by incubation on ice for 30 min. For each sample, 6 μL Protein A/G beads (Thermo Fisher, 88803) were blocked in PDB buffer with 5% skim milk powder for 1 h at 4 °C, washed in PDB and added to the protein solution in 100 μL PDB, followed by incubation for 1 h at 4 °C. Washes were performed twice with each of the following buffers: PDB buffer with 5% skim milk powder, high-salt PDB buffer (250 mM KCl), and PDB buffer with added 0.2% Sarkosyl and 5% skim milk powder. Final washes were performed once with 500 μL PDB buffer and once with 40 μL PDB. Proteins were eluted by incubating the beads at room temperature for 10 min in 15 μL of 1× Bolt LDS loading buffer (Invitrogen, B0007). After elution from beads, β-mercaptoethanol was added to a final concentration of 5%, and samples were heated at 70 °C for 10 min. Protein samples were examined by western blotting.

## Coimmunoprecipitation of whole cell extract

Protein A/G beads (Thermo Fisher, 88803) were washed in RIPA-IP 137 (137 mM NaCl, 50 mM Tris-HCl pH 7.5, 10% NP-40) and incubated in RIPA-IP 137 + 1% BSA for 30 min at 4 °C. Beads were then incubated in 1 μg of either anti-TOP1 (Abcam, ab109374), anti-MYC (Abcam, ab32072), or anti-IgG (Cell Signaling Technology, 2729) primary antibodies in RIPA-IP 137 for 3 h at 4 °C. HCT116 cells were harvested, spun down and washed once in PBS (Thermo Fisher, 14190169). Cells were then lysed by sonication using Bioruptor (Diagenode) at high amplitude with 20 cycles of 30 s ON and 30 s OFF in native lysis buffer (50 mM HEPES, 150 mM NaCl, 1 mM EDTA, 1% Triton X-100) supplemented with protease inhibitors (cOmplete, EDTA-free protease inhibitor cocktail (Merck, 4693132001), 0.5 mM AEBSF (Merck, A8456), 10 μM Leupeptin (Merck, L2023), and 1 μM Pepstatin A (Merck, P5318)), before treating with 12.5 units of benzonase (Merck, E8263) for 15 min. Next, cells were centrifuged to separate the cell debris from the resulting lysate. Lysate from 1 million cells was pre-cleared with blocked protein A/G beads without antibody for 1 h at 4 °C. Beads incubated with anti-TOP1 and anti-IgG antibodies were washed twice in RIPA-IP 137 before adding the pre-cleared lysate and incubating for 1 h at 4 °C. Supernatant from the resulting TOP1 IP was then incubated for 1 h with anti-MYC bound beads at 4 °C, while the supernatant from the IgG IP was incubated with another round of anti-IgG bound beads. Beads were washed twice with RIPA-IP 137, twice with RIPA-IP 137 0.2% sarkosyl (137 mM NaCl, 50 mM Tris-HCl pH 7.5, 0.2% sarkosyl) and twice with RIPA-IP 300 (300 mM NaCl, 50 mM Tris-HCl pH 7.5, 10% NP-40). All wash buffers were supplemented with the protease inhibitors. Proteins were eluted from beads at room temperature in 1× NuPage LDS sample buffer (Invitrogen, NP0007). After elution from beads, β-mercaptoethanol was added to a final concentration of 5% together with 0.2 M DTT, and samples were heated at 70 °C for 10 min. Samples were probed using western blotting as described above.

## Chromatin-bound protein harvest for western blot from Halo-TOP2A HCT116^TOP1-AID cells

Halo-TOP2A HCT116^TOP1-AID cells were treated with 0.1 mM auxinole for 24 h to block TOP1 degradation, or with 500 μM auxin for 60 or 120 min. Cells were counted using a Countess 3 cell counter (Invitrogen), and 1 × 10^6 cells were harvested per condition. For the TOP1 harvest in the chromatin fraction, the same protocol as for the glycerol gradients was followed.

## Single molecule tracking (SMT)

Halo-TOP2A HCT116, Halo-TOP2A HCT116^MYC-AID or Halo-TOP2A HCT116^TOP1-AID cells were plated in 4-well Lab-Tek II Chamber slides (Nunc, 155382) in phenol-free high-glucose DMEM media (Thermo, 31053028) with 10% FBS and GlutaMAX to achieve 40-50% confluence upon imaging 48 h later. To image nucleoli, we transfected with a 1:99 ratio of GFP-Nucleolin vector (Addgene, 28176) and an empty vector (pCDNA4/TO, Thermo, V102020) 24 h after plating using Lipofectamine 3000 (Thermo, L3000008), according to the manufacturer's protocol. The TOP2A was then labeled with 100 nM photo-activatable Janelia Fluor (JF) 549 Halo ligand and 10 nM JF-646 Halo ligand (both gifts from Luke Lavis) for 30 min at 37 °C in the cell culture incubator. The cells were then washed three times with warmed PBS and incubated for 15 min in growth media. This washing process was repeated once before the media was replaced again with growth media to remove all excess Halo ligand. If cells were treated with 500 μM auxin, the auxin was included during both 15 min incubations and in the final media suspension to ensure knockdown had occurred for at least 30 min for MYC degradation and 75 min for TOP1 degradation before imaging. Treatment with 5 μM triptolide (Merck, T3652), 10 μM camptothecin (Merck, 208925) or 2,5% 1,6-hexanediol (Merck, 240117) was added in the final solution and imaging would begin after the requisite amount of time.

Cells were imaged using a custom-built microscope able to perform single-molecule imaging by HILO illumination as previously described[32]. Nuclear TOP2A distribution was determined from the JF-646 channel, enabling distinction between the nucleolus and the nucleoplasm based on the staining intensity. Photo-activation of the JF-549 ligand was achieved by tuning the power of a 405 nm laser to ensure 2–5 molecules of JF-549 ligand were photo-active in each frame. For each video, 3000 frames were captured at 100 frames per second. Measurements for the nuclear, nucleolar, and nucleoplasmic regions were determined by only analyzing the region of interest after manually segmenting based on the TOP2A JF646 signal. The path of individual molecules from the SMT video was determined using the ImageJ Trackmate plugin[82] without gaps being allowed and with 0.5–1 μm maximum linking distance depending on track density. Tracks were classified into fast, slow, and bound molecules using the vbSPT algorithm[34], excluding tracks shorter than 4 frames.

To calculate diffusional anisotropy, we removed bound track segments and calculated the angle between consecutive displacements. The anisotropy index was calculated as the ratio of the number of backward jumps (with an angle between jumps in the [150°, 210°] range) over the number of forward jumps (angles in the range [−30°, 30°]). Plotting the anisotropy index as a function of the distance run by the molecule allows discrimination between exploration modes. Error bars were calculated as the standard deviation from 100 random subsampling composed of 50% of the original data. Analysis of diffusion coefficients and fractions at the single cell level was performed by analyzing the distribution of displacements with a three-component diffusion model, as described previously[32].

Cross-correlation analysis was performed as previously described[32]. Briefly, co-clustering between bound and diffusing molecules was evaluated by first identifying the centroid of each track segment corresponding to bound molecules, and then calculating the cross-correlation between these centroids and the positions of all the particles classified as either "slow diffusing" and "fast diffusing". The resulting cross-correlation plots were then compared to those obtained from the distances between bound molecules and random positions in the nucleus.

The spectrum of diffusion coefficients was evaluated from SMT data using SASPT, a method that uses a Bayesian nonparametric approach to infer the posterior distribution of diffusion coefficients across an array of finite possible states (diffusion coefficients ranging from $10^{-2}$ to $10^2$). SASPT was run with the following settings: focal depth 0.7 μm, sample size 10000, likelihood type RBME. Mean posterior occupation across the population of tracked cells is displayed[83].

## Quadruple-trap (Q-TRAP) optical tweezers

Single-molecule TOP2A decatenation assays were performed using the Q-TRAP optical tweezers (Lumicks). Two molecules of lambda DNA with biotinylated ends (Lumicks, 00001) were each bound to two SPHERO streptavidin-coated polystyrene particles (Spherotech, SVP-40-5) held by optical tweezers. These particles were manipulated to intertwine the two DNA molecules, forming five DNA crossovers. This structure was then moved into a channel that had been passivated overnight in passivation buffer (50 mM Tris pH 8, 150 mM NaCl, 10 mM MgCl$_2$, 0.5% pluronic, 0.1% BSA). This channel contained 10–50 nM SNAP-TOP2A labeled with SNAP-647 ligand, 100–300 nM MYC or equivalent BSA, 50 mM Tris pH 8, 150 mM NaCl, 10 mM MgCl$_2$, 450 nM BSA, 0.5 mM DTT, 50 µM Sytox Orange, and 2 mM ATP where required.

The exact time of entrance of the DNA crossover into the protein channel was determined by the background photon count in the TOP2A channel. The exact time of decatenation was determined by the change in the force acting on the polystyrene particles. Droplet formation at the crossover was measured by comparing the intensity of TOP2A at the DNA crossover, subtracting the background TOP2A signal surrounding the crossover and comparing this value to a threshold for TOP2A crossover binding.

## Stimulated emission depletion (STED) microscopy

HCT116 cells were plated onto 18 mm coverslips (Marienfeld, 630-2200) in 12-well plates and grown to 50-60% confluency. After the indicated treatment, cells were fixed in 3% paraformaldehyde (PFA, Pierce, 28908) in PBS for 5 min at room temperature. After washing with PBS, cells were permeabilized in methanol for 3 min at −20 °C, before washing again in PBS and blocking in 2% BSA (Merck, A7906) in PBS for 30 min at room temperature. Proteins were fluorescently labeled by sequential incubation with primary (1:200 dilution) and fluorescently labeled secondary (1:400 dilution) antibodies for 1 h each in 0.5% BSA/PBS. These antibodies include anti-MED4 (Thermo Fisher, 67839-1-IG), anti-H3.3 (Active Motif, 91191-AF), anti-RPA194 (Santa Cruz, sc-46699), anti-nucleophosmin (Abcam, ab10530), anti-TOP2A (Abcam, ab52934), anti-mouse-AF555 (Thermo Fisher, A-21422), and anti-rabbit-647 (Thermo Fisher, A-21245). For triple protein labeling, the rabbit anti-fibrillarin conjugated to Alexa Fluor 488 (Abcam, ab184817) was used. To prevent binding of this antibody to other rabbit secondary antibodies, following the secondary antibody incubation and subsequent wash, the slides were incubated with rabbit IgG (Merck, 12–370, 1:200 dilution) to block free anti-rabbit secondary antibody binding sites. Then, the slides were incubated with anti-fibrillarin-488 (1:200), with DAPI if required. After washing, coverslips were mounted onto slides in a hard-setting ProLong Diamond Antifade Mountant (Thermo, P36970). In some cases, DAPI was not included to prevent bleed-through into other channels, since the nucleus was clearly visible by TOP2A/MED4 staining.

Multi-color STED imaging was performed on a Leica SP8X system, equipped with a tunable white-light laser (excitation 470–670 nm), high-power green/orange/red depletion lasers (592, 660, 775 nm), a chromatically optimized 100×/1.4 STED WHITE oil immersion objective, and single photon sensitive detectors (HyD-SMDs). Nuclear protein structures were imaged sequentially frame by frame (1024 × 1024 pixels) at a speed of 200 lines per second (4-line averages) with a pixel size of 25 nm and pinhole settings of 0.9-1.0 Airy units. STED images were deconvoluted using the Huygens 24.10 software (Scientific Volume Imaging) before data analysis. Images were analyzed using custom Matlab scripts. Briefly, the central points of nucleoplasmic puncta were determined at a sub-pixel resolution. Then, the signal from the observed channel surrounding the detected puncta is averaged and plotted.

## Airyscan microscopy

HCT116 cells were plated onto 18 mm coverslips (Marienfeld, 630–2200) in 12-well plates and grown to 50% confluency and treated with 5% 1,6-hexanediol for 5 min. After treatment, cells were fixed, permeabilized, blocked, and labeled for TOP2A and MED4 using the materials and methods as described for STED imaging. Cells were imaged using a Zeiss LSM 900-Airy confocal microscope using the 63× magnification objective.

## Quantitative PCR (q-PCR)

HCT116 cells were treated as indicated, then RNA was purified using NucleoSpin RNA columns (Macherey-Nagel, 740955) according to kit instructions. Equal amounts of RNA resuspended in 13 ul H$_2$O were mixed with 1 ul of 10 mM dNTP (10 mM each of dATP, dCTP, dGTP, dTTP; Promega, U1205, U1215, U1225, U1235) and 0.5 ul of random hexamers (Promega, C1181). This mixture was heated at 65 °C for 5 min, then put on ice for 1 min. 4 ul RT buffer, 0.3 ul SuperScript IV Reverse Transcriptase (both Thermo, 18090200), 1 ul 0.1 M DTT (Merck, 43815) and 0.2 ul of RNAsin Ribonuclease inhibitor (Promega, N2515) were added. The solution was incubated at room temperature for 10 min, followed by 50 °C for 30 min to transcribe cDNA, and at 80 °C for 10 min to stop the reaction. The cDNA was diluted, and quantitative PCR was performed using Fast SYBR Green Master Mix (Thermo, 4385612) with previously published[84] primers for the internal transcribed sequence (ITS) of the 47S pre-ribosomal RNA (F: CCG TGG CCT TAG CTG CTC GC, R: CCC ACT TAA CTA TCT TGG GCT G) and beta-2-microglobulin (B2M; F: GAA ACC TTC CGA CCC CTC T, R: GCC AGA CGA GAC AGC AAA C). The ΔΔCt method was used to measure ITS expression relative to B2M, normalized to the control condition.

## TOP2 covalent adduct detection coupled to sequencing (TOP2 CAD-seq)

$3 \times 10^7$ (HCT116$^{MYC-AID}$) cells were treated (in biological triplicates) with auxin, followed by 10 µM MG132 (Merck, 474790) for 30 min and 100 µM etoposide in the last 6 min to trap TOP2cc. Cells were lysed in 4.5 mL of M buffer (9.3 mM Tris-HCl pH 6.5, 18.6 mM EDTA, 5.59 M guanidine thiocyanate, 0.93% DTT, 0.93% Sarcosyl, 3.72% Triton X-100) and briefly sonicated with Bandelin probe sonicator at 20% amplitude for 1 s. DNA covalent adducts were precipitated with 50% EtOH at −20 °C, centrifuged at 21130 × g at 4 °C and pellets were washed thrice in wash buffer (20 mM Tris-HCl, pH 7.5, 50 mM NaCl, 1 mM EDTA, 50% EtOH). Ethanol leftovers were aspirated, and pellets were dried for 5 min and resuspended in 1 ml of EB-SDS 0.05% (10 mM Tris-HCl, pH 8, 0.05% SDS). After 30 min incubation by gentle agitation, 500 µl of each sample was further fragmented with benzonase (0.02 U/µg) at 37 °C for 15 min. The reaction was stopped with 50 mM EDTA and 0.1% SDS. For the immunoprecipitation 3.5 µg of a 1:1 mixture of anti-TOP2A (sc166934 and ab52934) antibodies were mixed with 25 µl Protein A/G magnetic beads (Pierce, 88803) and incubated at 4 °C for 4 h with rotation. Beads were washed once with ice-cold PBS, and DNA covalent adducts from $1.5 \times 10^7$ cells were added to the Protein A/G-antibody complexes and incubated overnight at 4 °C with rotation. Samples were washed once with RIPA buffer, once with RIPA buffer containing 300 mM NaCl, once with LiCl-SDS 0.1% buffer (10 mM Tris-HCl pH 8, 1 mM EDTA pH 8, 250 mM LiCl, 0.5% NP40, 0.5% Na-Deoxycholate, 0.1% SDS) and twice with TE-SDS 0.1% (10 mM Tris-HCl pH 8, 1 mM EDTA pH 8, 0.1% SDS). The beads were then resuspended in 100 µl TE plus 0.5% SDS supplemented with proteinase K (500 µg/ml) and incubated for 4 h at 60 °C. Samples were then purified using the QIAquick PCR Purification Kit.

## Library preparation and sequencing of TOP2 CAD-seq

DNA from TOP2A CAD samples was quantified with the Qubit dsDNA HS Assay Kit (Thermo Fisher, Q33230). To cleave off covalently bound tyrosyl residues from TOP2A, the samples were additionally treated

with ExoVII (NEB, M0379S) (0.5 U per 10 ng of DNA) and purified by PCR Purification Kit (QIAGEN, 28106). Sequencing libraries were created according to the ThruPLEX DNA-seq kit protocol (Takara, R400676). Size selection was performed in the range of 200–700 bp with AMPure XP beads (Beckman, A63880) and confirmed using the Agilent High Sensitivity DNA Kit (Agilent, 5067–4626) on the Agilent 2100 Bioanalyzer. Libraries were pooled and sequenced using the NextSeq 1000/2000 P2 XLEAP-SBS Reagent Kit (100 Cycles) (Illumina, 20100987). The sequencing run was Pair End and Dual Index with 2 × 50 bp reads.

## SLAM-seq

A modified version of SLAM-seq[85] was used to measure nascent RNAs over the full transcript length. A total of $2 \times 10^6$ HCT116[MYC-AID] cells were treated (in biological triplicates) with 500 μM auxin for 90 min or with 0.1 mM auxinole for 24 h before harvesting. 4-Thiouridine (S4U; 100 μM) was added to the medium of all samples in the dark 75 min before the harvest. Cells were washed twice with fresh medium and scraped stepwise in 2 × 0.5 ml of ice-cold TRIzol LS Reagent (Invitrogen, 10296010) and proceeded immediately to the next step. For RNA isolation, we followed the manufacturer's instructions of SLAM-seq Kinetics Kit-Anabolic Kinetics Module (Lexogen, 061.24) except for the following modifications: Samples were incubated for 5 min at 65 °C and mixed with 200 μl of chloroform:isoamyl alcohol mix (24:1) (Merck, 25666), before further incubation for 15 min at 65 °C under strong agitation (2000 rpm) and repeated vortexing. RNA in the aqueous phase was separated by centrifugation at 16000 g for 15 min at 4 °C, supplemented with the provided reducing agent and again extracted with one amount of chloroform:isoamyl alcohol mix (24:1) before precipitation overnight at −80 °C. RNA was pelleted for 20 min at $16000 \times g$, washed twice with 1 and 0.18 ml of 80% EtOH (plus reducing agent), respectively, and pelleted for 10 min at $7500 \times g$. Resuspension in the elution buffer (provided by the SLAM-seq Kinetics Kit) was facilitated by incubation for 15 min at 55 °C, 10 min on ice and gentle pipetting to avoid fragmentation of RNA. RNA was quantified with the Qubit RNA HS Assay Kit (Invitrogen, Q32852) and its integrity was controlled using the RNA 6000 nano kit (Agilent, 5067–1511) on a Bioanalyzer 2100 (Agilent). To facilitate normalization of the sequencing reads, the isolated RNA was mixed with ERCC RNA Spike In-Mix (Invitrogen, 4456740). Specifically, RNA (4.5 μg) was mixed with 2 μl of 1:10 diluted ERCC RNA Spike In-Mix, and the elution buffer was added to fill up to 10 μl. Next, 4.5 μg RNA mixed with ERCC was taken, the elution buffer was added up to 15 μl, and alkylation was performed by the addition of iodoacetamide following the manufacturer's instructions. Once again Qubit RNA HS Assay Kit (Invitrogen, Q32852) was used to quantify the RNA, and RNA integrity was controlled using the RNA 6000 nano kit (Agilent, 5067–1511) on a Bioanalyzer 2100 (Agilent). Ribosomal RNAs (rRNAs) were depleted from 950 ng of alkylated RNA using the RiboCop (HMR) kit (as part of the CORALL Total RNA Seq V2 12nt B1 UDI12B_0001-0024, Lexogen, 184.24) following the manufacturer's instructions, except for increasing the volume for RNA elution to 16 μl. Removal of rRNAs was confirmed using the RNA 6000 Pico Kit (Agilent, 5067–1513) on a Bioanalyzer 2100 (Agilent).

## Library preparation and sequencing of SLAM-seq

To prepare cDNA libraries spanning the whole transcript length, the CORALL Total RNA Seq V2 12nt B1 UDI12B_0001-0024 (Lexogen, 184.24) was used, which includes reverse transcription with displacement stop primers. Ten microliters of the rRNA-depleted RNA were used as input material, and the final library was amplified by 15 PCR cycles as defined from the PCR Add-on and reamplification Kit V2 for Illumina (Lexogen, 208.96). DNA concentration and molarity were determined by Qubit dsDNA HS Assay Kit (Invitrogen, Q33231) and Bioanalyzer High Sensitivity DNA Kit (5067–4626) on a Bioanalyzer 2100 (Agilent), respectively. Libraries were pooled and sequenced

using the NextSeq™ 1000/2000 P2 XLEAP-SBS™ Reagent Kit (100 Cycles) (Illumina, 20100987). The sequencing run was paired-end and dual index with 2 × 53 bp reads.

## TOP2 CAD-seq data analysis

The generated fastq files were quality controlled with FastQC (https://www.bioinformatics.babraham.ac.uk/projects/fastqc/) and MultiQC[86], trimmed with cutadapt (http://journal.embnet.org/index.php/embnetjournal/article/view/200/479), aligned to the hg38 reference genome with bowtie2[87], deduplicated, sorted and indexed using Samtools[88] and Picard (http://broadinstitute.github.io/picard). RPM normalized BigWig files were generated with bamCoverage[89]. The coverage matrices, profiles and heatmaps of short reads' average distribution near TSSs and along normalized gene bodies of protein-coding genes were generated by deeptools commands computeMatrix and plotHeatmap. The expression of the genes was determined based on the SLAM-seq data[12]. The genes were ranked according to SLAM-seq spike-in normalized counts in descending order, and the top 1500 were selected for Fig. 5G, H. Peaks were called using Macs3 callpeak, and peak summits were used for plotting.

## SLAM-seq data analysis

Nascent RNA reads from the SLAM-seq were processed using the established pipeline slamdunk[90] (https://github.com/jkobject/slamdunk). Since the sample preparation deviated from the standard SLAM-seq method, for instance, in the library preparation, the pipeline was adapted accordingly. The quality of the FASTQ files was assessed with FastQC (https://www.bioinformatics.babraham.ac.uk/projects/fastqc/) and MultiQC[86]. UMI extraction was performed as in the CORALL analysis pipeline from Lexogen (https://github.com/Lexogen-Tools/corall_analysis) and adapter trimming of FASTQ using cutadapt (v.5.2, http://journal.embnet.org/index.php/embnetjournal/article/view/200/479). Read were aligned with NextGenMap (v0.5.5, https://github.com/Cibiv/NextGenMap) to the hg38 and ERCC reference genome, and the resulting files were sorted and indexed using Samtools[91]. After deduplication with UMI_tools[92] (v1.1.6), slamdunk filter was run with hg38 expressed protein-coding genes or ERCC genes as opposed to just the 3′ untranslated regions, followed by single-nucleotide polymorphism detection via slamdunk snp. Nascent human reads with T > C conversions were counted using slamdunk count using a T > C conversion threshold of ≥ 2. Then, T > C conversions were separated from background reads using the alleyoop read-separator command from the slamdunk package. Human T > C counts were normalized to ERCC filtered reads for each sample by applying a spike-in-based scaling factor defined as (ERCC filtered reads in the lowest-ERCC sample) / (ERCC filtered reads in each sample). Spike-in normalized T > C counts per gene were then divided by gene length (kb) and $\log_{10}$-transformed after addition of a pseudocount (+1).

## Declaration of generative AI and AI-assisted technologies in the writing process

During the preparation of this work, the authors used ChatGPT o1 in order to improve the readability of the manuscript. After using this tool/service, the authors reviewed and edited the content as needed and take full responsibility for the content of the published article.

## Statistics and reproducibility

All image analysis for nuclear localization of TOP2A, γH2AX, and droplet imaging experiments (for 2 μM and 0.5 μM concentrations) was done using CellProfiler version 4.2.8, and statistical analysis was performed in GraphPad Prism version 10.4.1. P-values are shown only when comparisons are significant ($p < 0.05$). Western blot analysis was performed using Image Lab Version 6.1.0 build 7, and plots, and statistical analysis was performed on GraphPad Prism. No statistical method was used to predetermine sample size. No data were excluded

from the analyses. The experiments were not randomized. The Investigators were not blinded to allocation during experiments and outcome assessment. All cell lines used are available on request from Laura Baranello.

### Reporting summary

Further information on research design is available in the Nature Portfolio Reporting Summary linked to this article.

## Data availability

The data for this study are deposited in the Gene Expression Omnibus (GEO) with accession number GSE311618. Original data are available through Figshare at https://doi.org/10.6084/m9.figshare.29467730. Previously published ChIP-seq data can be accessed in GEO with the accession numbers GSE241338[19] and GSE181450[5]. Source data are provided with this paper. All additional information and data reported in this paper are available from the corresponding author upon request. Source data are provided with this paper.

## Code availability

All code related to STED imaging analysis and pixel correlation visualization can be found at GitHub (https://github.com/don-cameron/Cameron-et-al.−2025/)[93]. Any further information required to reanalyze the data in this manuscript can be requested from the corresponding authors.

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

## Acknowledgments

This work was supported by the ERC Consolidator Grant (project no. 101088643 to L.B.), Knut och Alice Wallenbergs Stiftelse (KAW 2022.0380 and KAW 2022.0189 to L.B.), the Swedish Research Council (2021-02630 and 2024-02583 to L.B.), Cancerfonden (21 1771 Pj 01 H and 24 3815 Pj 01 H L.B.), and KI Consolidator (2-190/2022 to L.B.). This work was supported by an ERC Consolidator Grant (project no. 866238) and the Swedish Research Council (2020–03400) to F.W. E.P. acknowledges funding from the Area of Advanced Nano at Chalmers. D.M. acknowledges funding from Worldwide Cancer Research (Grant Reference number 22-0116) and from AIRC under the IG 2018-ID:21897, IG 2023-ID:28792. V.L. acknowledges the support of the Integrated Structural Biology platform of the Strasbourg and Instruct-ERIC center IGBMC-CBI, supported by FRISBI (ANR-10-INBS-0005), and funding from the Inter-disciplinary Thematic Institute IMCBio +, supported by IdEx Unistra (ANR-10-IDEX-0002), and by the SFRI-STRAT'US project (ANR-20-SFRI-0012) under the framework of the France 2030 Program. B.L.D. is a doctoral fellow supported by EUR IMCBio (ANR-17-EURE-0023). D.P.C. was supported by AMED-ASPIRE-A (20H05933), AMED BINDS (22ama121020j0001), the Helge Ax:son Johnson Foundation (F22-0197), the Erik and Edith Fernström Foundation for Medical Research (2022-00603) and the UTokyo-KI LINK program. We thank the SciLifeLab/NMI Advanced Light Microscopy unit for STED support. The computations and data storage were enabled by resources in project [SNIC 2018/8–390], provided by the Swedish National Infrastructure for Computing (SNIC) at UPPMAX, partially funded by the Swedish Research Council through grant agreement no. 2018-05973. Part of this work was facilitated by the Protein Science Facility at Karolinska Institutet/SciLifeLab (https://ki.se/psf) and the Karolinska Genome Engineering Facility (KGE). Cell imaging was performed at the Biomedicum Imaging Core (BIC) with support from Karolinska Institutet. We thank Dr. Johannes Zuber, Dr. Yves Pommier, and Dr. Shar-Yin Huang for sharing reagents and helpful advice. We thank Dr. Camilla Björkegren, Dr. Kristian Jeppsson, Dr. Arne Lindqvist, Dr. Fedor Kouzine, Dr. Keir Neuman, Dr. Katsuhiko Shirahige, and Dr. David Levens for helpful advice and critical discussion. We thank Dr. Katsunori Fujiki, Dr. Florian Salomons, Dr. Hans Blom, Dr. Ola Larsson, Dr. Solenne Bleuse, and Dr. Claudia Kutter for sharing instruments and technical expertise. The photo-activatable JF549 Halo-tag was generously gifted by Dr. Luke Lavis.

## Author contributions

Authors with equal contributions are listed in alphabetical order. Conceptualization: D.P.C., K.J., D.M., and L.B. Methodology: D.P.C., K.J., A.L., C.M., V.K., E.I., M.M., A.P., F.W., D.M., and L.B. Software: D.P.C., K.J., C.M., V.K., E.I., and D.M. Formal Analysis: D.P.C., K.J., C.M., V.K., E.I., and D.M. Investigation: D.P.C., K.J., A.L., C.M., V.K., M.M., E.I., H.K., A.P., E.P., B.J., and B.S.L.D. Resources: V.L., F.W., D.M., and L.B. Writing - Original Draft: D.P.C., K.J., and L.B. Writing - Review and Editing: D.P.C., K.J., A.L., C.M., V.K., M.M., E.I., E.P., B.J., B.S.L.D., V.L., F.W., D.M., and L.B. Visualization: D.P.C., K.J., C.M., V.K., E.I., and L.B. Supervision: D.P.C., V.L., F.W., D.M., and L.B. Funding acquisition: D.P.C., V.L., F.W., D.M., and L.B.

## Funding

## Competing interests

The authors declare no competing interests.
