## [Transparent Peer Review file · Nature Communications]

MYC modulates TOP2A diffusion to promote substrate detection and activity

Corresponding Author: Dr Laura Baranello

Version 0:

Reviewer comments:

Reviewer #1

(Remarks to the Author)

The manuscript by Cameron and colleagues studies the intranuclear trafficking of TOP2A, its partitioning between the nucleolus and the nucleoplasm, its diffusion, its association with MYC, and its relationship to transcription. The data, the results, and discussion are interesting and generally well-done, and they suggest the importance of regulating intranuclear diffusion. Nevertheless, there are several major and minor issues that if addressed would improve the presentation so that it would merit presentation to the broad community of biophysicists, biochemists, and cancer biologists.

Major issues:

Previous work from this group has shown that MYC combines with TOP2A and TOP1 in a “topoisome”. The current work does not provide any context for discriminating the actions and trafficking of a TOP2A-MYC complex versus a topoisome. (Incidentally it is unclear to this reviewer if the previously defined topoisome is the same as the currently described TOPOisome. If they are the same, then the previous term “topoisome” should be retained for simplicity. As the previous term has been used already in reviews and other papers, there is no need to complicate the terminology by introducing a new term.) The authors focus on diffusion by TOP2A-MYC complexes and do not address the issue of distinguishing these bimolecular complexes from topoisomes. What are the relative abundances of TOP2A-MYC versus topoisomes? If TOP1 were to be doped into the TOP2A condensates in vitro, how would the rates of diffusion vary? How would the addition of a TOP1 poison (i.e. CPT) influence the diffusion rate of TOP2A?

Line 283. It is not clear to me that condensates are forming on the ionically immobilized-DNAs rather than polymer engagement TOP2A molecules. The authors assert that because the sizes of spots vary, proves that multiple TOP2As must be included—this assertion doesn't hold water in my view. The sizes of the interstices of the tangled DNA molecules that engage a single TOP2A would be expected to vary considerably, and if so then, so to would the excursions of the trapped molecules vary accordingly. But, if the number of molecules per cage varied, then the brightness would be expected to change incrementally. So, the relative immobility of the TOP2A and its occurrence at very concentrations may not be a result of phase separation but may reflect entrapment within a DNA matrix due to a high local concentration of binding sites.

Line 394. The authors claim that there is no general downregulation of transcription after auxin-depletion of MYC. Therefore, the loss of TOPSAcc's at promoters was not due to decreased transcription. But they make this assessment using whole cell fluorescence after EU incorporation. At the cellular level rRNA accounts for the vast majority of RNA synthesis. So, without some measure of ongoing RNAPII activity, the authors cannot be certain that transcription is unimpaired. In fact, wouldn't ongoing transcription be expected to decline when MYC is degraded?

Minor issues:

Line 84: co-localization, yes, but on what basis is this a condensate?

Line 234: relieve, not relief

Line 247: The authors have an interesting result upon which they make no comment. Triptolide actually increased TOP2A mobility in all cases. This should be mentioned. Does the inhibition of ERCC3 by the drug inhibit PIC formation or does it release previously recruited TOP2A?

Line 262 how does this compare with in vivo concentration?

(Remarks on code availability)

Reviewer #2

(Remarks to the Author)

Topoisomerases are essential proteins that resolve DNA supercoiling and entanglements, thereby relieving topological stress. In a previous study, the authors' group showed that the oncoprotein MYC stimulates topoisomerases to remove DNA entanglements generated by oncogenic transcription. In the present study, the authors claim that TOP2A in human cells exists in a dynamic equilibrium between sequestration in the nucleolus, substrate searching in transcription condensates, and active engagement on chromatin (three distinct diffusion states). Using single-molecule approaches, they suggest that MYC regulates TOP2A activity by affecting the diffusion state of TOP2A-containing complexes ("TOPoisomes"). Interestingly, MYC accelerated the diffusion of TOP2A in the nucleoplasm, potentially by limiting the size of protein complexes containing TOP2A. This may enhance TOP2A's substrate-searching ability, thereby increasing its activity at transcribed regions. While the authors' story is intriguing, several points are missing to support their main conclusion, and some aspects of data interpretation and presentation are also weak. My specific comments are as follows.

Major points:

- 1) Fig. 1. As the authors mention, TOP2A expression increases during S-phase, peaks in mitosis, and is subsequently degraded (Refs. 48, 49). The nucleolar accumulation of TOP2A during the cell cycle should be presented. In addition, did the total TOP2A level in the nucleus change after either TOP1 depletion or ICRF treatment? It is important to rule out the possibility that the decrease in the nucleolar/nucleoplasmic ratio is due to changed TOP2A levels rather than translocation.
- 2) Fig. 2I–J. Although the authors claim that the slowly diffusing TOP2A corresponds to molecules diffusing in transcriptional condensates, direct evidence appears to be lacking. The data only show that the TOP2A inhibitor ICRF-193 reduces the accumulation of TOP2A at MED4 puncta, whereas the slow fraction remains unchanged following ICRF treatment (Fig. 2C).
- 3) Fig. 5E. I wonder whether TOP2A can form liquid droplets under the buffer conditions used in the Q-trap assay. The buffer used in the droplet assay contained 20% glycerol, which may be required for TOP2A–DNA droplet formation. The buffer for the Q-trap assay, however, did not contain glycerol. If TOP2A does not form droplets under the Q-trap assay conditions, how can MYC accelerate the binding events of TOP2A at DNA crossovers?
- 4) Overall, the quality of data presentation is low. For instance, in Fig. 2I and Extended Data Fig. 3A, are the scale bars accurate? The TOP2A puncta appear to be around 5 μm in size, which seems too large.
- 5) Extended Data Fig. 6g. This figure indicates that the cell population with low EU increased after MYC depletion. This appears to conflict with the authors' interpretation (Line 396).

Minor points:

- 1) Fig. 1A. This panel is confusing. DAPI staining is needed to visualize the nuclear structure.
- 2) Line 95. It is known that RNAPII transcription inhibitors alter the nucleolar structure (nucleolar cap formation).
- 3) Which MYC subtype did the authors use? Please specify in the Methods section.
- 4) Line 222. Structured illumination light sheet microscopy (SIM)?
- 5) Code availability: The provided codes (<https://github.com/don-cameron/Cameron-et-al.-2025/tree/main>) have empty "readme.md" files for pixel correlation plotting and for the general instructions for the entire dataset!

(Remarks on code availability)

Reviewer #3

(Remarks to the Author)

(Remarks on code availability)

I co-reviewed the code with one of the reviewers who provided the listed reports.

Reviewer #4

(Remarks to the Author)

This is very interesting and strong work built on the previous finding from some co-authors as well as an existing literature. It provides a step forward in the understanding of MYC function which is a big and important question in Multiple fields. Study used an impressive set of biophysical, biochemical and cell-based methods.

Below is my review:

The manuscript of Cameron and co-authors uncover one of the mechanisms of how MYC boosts transcription. They provided data that TOP2A exists in cells in a balanced distribution between nucleolar and nucleoplasm. In nucleoplasm, TOP2A is distributed between different states depending on its diffusion coefficients: a bound fraction interacting or catalytically engaged with chromatin, a slow fraction enriched in transcription condensates, and a fast fraction diffusing through the nucleoplasm. They then showed that degradation of MYC reduced TOP2A diffusion rates in slow and fast fractions. They showed that MYC promotes TOP2A mobility via reducing the size of TOP2A droplets in cell free conditions and TOP2A complexes in cells. This allows faster search of targets by TOP2A and more efficient resolution of supercoiling needed for transcription.

This is very strong and rigorous study using multiple experimental approaches, as well as accurately and clearly written. I just have several questions/comments, which I hope authors can clarify.

Lines 78-79: "was lost along with nucleolar protein fibrillarin upon further treatment with RNase A (Fig. 1b and Extended Data Fig. 1d), suggesting its association with nucleolar structures was maintained by ribosomal RNA". Was the mechanism of TOP2A association with rRNA already known? Please provide refs or explain.

Contradictory impression between image Fig 1B vs quantitation Fig. S1D. Control image shows signal from nucleolus and nucleoplasm, CSK image – only nucleolar signal, RNase image – only nucleoplasm. Quantitation shows that control = RNase. Please explain.

Figure 1E and F and all other plots of the same type show very few dots while more around thousand cells were measured and mentioned in the legend. What do dots represent?

Lines 123 – 125: "We observed an increase in DNA damage only after TOP1 degradation by auxin and etoposide treatment" and Fig.1F, ext Fig.2A and B. It is unclear why the gammaH2AX signal increased only after TOP1 degradation. Etoposide interactions with TOP2A should be independent on TOP1. Also based on presented images – without TOP1 degradation, it is decreased in response to etoposide treatment in ext fig 2A, and increased in ext fig 2b.

Lines 151-153: "Notably, TOP2B distribution across the nucleus was much more homogeneous as compared to TOP2A, although ICRF-193 also caused a slight re-localization nucleolus-to-nucleoplasm (Extended Data Fig. 2f,g) suggesting it may also be responsive to similar stimuli to TOP2A, although the TOP2A response was more pronounced." Is there enough structural difference between two topoisomerases to explain why this mechanism is much more pronounced in TOP2A vs TOP2B. It may be good to discuss.

Line 163 "we found that nuclear TOP2A diffusion could be categorized into three distinct populations." It would be good to show the actual distribution of diffusion coefficients, like histogram or in some other way.

Figure 2G is unclear. Should it contain colored lines according to the colors of FAST and SLOW? What exactly is the black line?

Line 198: "If the slow TOP2A fraction reflects TOP2A localized in transcription condensates and the bound fraction corresponds to TOP2A interacting with chromatin," Should not the slow and bound overlap, since bound should perform its activity within transcriptional condensates.

"transcribed chromatin labelled with H3.3" – H3.3 is not a very specific mark of transcription, since although in general it is enriched in transcribed regions, these transcribed regions are not transcribed all the time, taking into account transcription bursts, and transcription of many genes in a time-dependent manner (e.g., cell cycle). More accurate marks of the presence of transcription can be used, like EU incorporation.

Line 242: "we found that although the proportion of TOP2A found in the fast, slow, and bound fractions were largely unchanged"... do you mean the amount or just the ratio between fractions? Then, was the amount changed?

"MYC mutant containing only residues 200-320 (MYC200-320)" why this form of myc is important? It was mentioned later that it does not bind TOP2A, but it should be better to explain at the first instance.

Fig.4H "average of six experiments": what four data points represent?

Why did they use K562 MYC-AID cells in just one section of the manuscript if they already made HCT116 MYC-AID cells?

Fig. 5A: Was the exposure for the control and auxin the same? It looks like the amount of TOP2A in the auxin condition has been increased. Some of the replicates in the extended data provide a better illustration of the author's interpretation.

Crossover experiment: what was visualized and quantified? Individual TOP2A molecules or droplets.

Fig. 5H: Does degradation of MYC lead to reduced transcription? Total EU incorporation was not reduced, but gene expression is a minor fraction of total transcription. How about these 1500 genes?

Line 461-463: "TOP2A molecules decrease by approximately 10%-15%. According to the Stokes-Einstein relationship, in first approximation, diffusion coefficients would scale as the cubic root of the diffusing complex, thus corresponding to roughly 30% to 50% larger complexes upon MYC degradation". Math here is unclear. Cubic root of 30 and 50 are around 3 and 3.6 and not 10-15.

(Remarks on code availability)

Version 1:

Reviewer comments:

Reviewer #1

(Remarks to the Author)

The authors have done a scholarly job responding to all of the reviews. I have no further criticisms or suggestions.

(Remarks on code availability)

Reviewer #2

(Remarks to the Author)

The authors have addressed most of my comments, and the revised manuscript has been considerably improved. There are a few remaining issues that should be addressed. After that, the paper will be ready for publication in Nature Communications.

1. Extended Data Fig. 3a–c: I wondered whether these data include the slow and bound particles localized in nucleoli, since most of the slow/bound particles were located in nucleoli, which are not linked to transcriptional condensates.
2. Fig. 2a: Representative movies of the single-particle tracking data should be presented to aid the readers' understanding.

(Remarks on code availability)

Reviewer #3

(Remarks to the Author)

(Remarks on code availability)

I co-reviewed the codes with one of the reviewers who provided the listed reports.

Reviewer #4

(Remarks to the Author)

Authors addressed all my comments and questions and I do not have any further requests. I am very impressed with their additional work.

(Remarks on code availability)

made.

Response to Reviewers

We greatly appreciate the helpful comments and suggestions from the three Reviewers. We have carefully considered each comment and taken comprehensive actions to address them. These comments prompted us to conduct many new experiments and perform several new analyses. The results of these experiments and analyses have provided new insights and further strengthened the original conclusions of the manuscript. Below, we start with a summary of significant changes to the main Figures and Extended Data Figures. This is followed by a detailed, point-by-point response to each comment from the three Reviewers, with each comment (red) followed by our response (black). In the resubmission, we have included a version of the manuscript where the changes are indicated in red.

Summary of Significant Figure Changes:

1. Reviewer #1 requested a clearer distinction between the topoisome and the TOP2A-MYC complex in terms of relative abundances, as well as clarifications of TOP1's contribution to TOP2-MYC dynamics. In the past 3 months we have worked intensely on this comment. Through co-IP experiments, we now estimate that approximately 10% of TOP2A interacts with MYC as part of the topoisome, while 4% forms distinct TOP2A-MYC complexes (**Extended Data Fig. 4c-e**). We also performed droplet assays to investigate the colocalization of TOP1 with TOP2A and MYC and to determine whether TOP1 influences TOP2A droplet behavior. We had not previously worked with TOP1 in these assays, therefore enabling these experiments first required the purification and characterization of human recombinant SNAP-tagged TOP1. We found that TOP1 colocalizes with the TOP2-MYC droplets and that MYC reduces droplet size both in the presence and absence of TOP1 (**Fig 4g,i and Ext Data Fig. 5m**). Lastly, we performed SMT experiments to probe the effects of TOP1 depletion (by CPT or by auxin-mediated acute degradation) on TOP2A diffusion. The results (shown in **Extended Data Fig. 4f-h**) indicate that inhibition or loss of TOP1 increases TOP2A diffusion. Since the TOP2A-MYC interaction persists without TOP1 (**Extended Data Fig. 4c,e**), these results support the conclusion that MYC limits TOP2A complex size to promote diffusion.
2. Reviewer #1 requested stronger evidence for Mediator and TOP2 enrichment in transcriptional condensates, and Reviewer #2 questioned whether the slow-diffusing fraction of TOP2A corresponds to molecules within transcriptional condensates. Regarding the latter comment, direct SMT co-tracking is technically unfeasible, as intranuclear SMT is performed in a sparse labelling regime, strongly reducing the probability of observing correlated motion

between two differently labelled proteins. However, we now provide additional evidence linking TOP2A to Mediator-containing condensates, and that those are likely associated with the TOP2A slow fraction. Specifically, 1,6-hexanediol (HD), which disrupts Mediator-containing transcriptional condensates¹, triggers TOP2A relocation to the nucleolar compartment (**Extended Data Fig. 1d**) and reduces the persistence of TOP2A puncta in the nucleoplasm (**Fig. 1d,e**). Consistent with these microscopy results, HD treatment also weakens the spatial cross-correlation between slow-diffusing and bound TOP2A molecules, with a two-folds reduction relative to the control, indicative of decreased proximity. These new results are now shown as part of **Extended Data Fig. 3a-c**.

3. Reviewers #1, #2, and #3 requested more quantitative assessment of transcriptional changes following MYC degradation. We therefore quantified nascent transcript levels across the genome using SLAM-seq² after 90 minutes of auxin-induced MYC degradation. This short-term depletion had essentially no effects on the transcriptional output of the cells, nor did it alter the transcription levels of the top 1500 transcribed genes (**Extended Data Fig. 6 g,h** and **Fig.5 h**). These findings further strengthen the interpretation that the changes in TOP2A activity result from direct regulation by MYC rather than secondary transcriptional effects (**Fig. 5g**).
4. Reviewer #1 suggested that the TOP2A puncta form as a result of polymer entrapment, where multiple TOP2A molecules could be connected by binding the same DNA molecule. We ruled out this possibility by showing that TOP2A puncta are observed independently of DNA, indicating that their formation does not rely on DNA bridging (**Extended Data Fig. 5d**).
5. Reviewer #3 suggested labelling newly transcribed RNA with EU instead of using H3.3 staining. While we agree that this approach is, in principle, more appropriate, we identified several technical limitations that, in our view, make it unsuitable for inclusion in the manuscript. Please see our discussion of this point and **Rebuttal Figure 3**.
6. We have included new control experiments, refined quantitative analyses, and corrected several inaccuracies in data presentation—particularly within the microscopy datasets—to improve robustness and clarity of the results.

Reviewer #1 (Remarks to the Author):

The manuscript by Cameron and colleagues studies the intranuclear trafficking of TOP2A, its partitioning between the nucleolus and the nucleoplasm, its diffusion, its association with MYC, and its relationship to transcription. The data, the results, and discussion are interesting and generally well-done, and they suggest the importance of regulating intranuclear diffusion. Nevertheless, there are several major and minor issues that if addressed would improve the presentation so that it would merit presentation to the broad community of biophysicists, biochemists, and cancer biologists.

We thank Reviewer 1 for their positive appraisal of our work. We have designed and performed a range of experiments which we believe addresses their issues, and we believe that this has significantly improved the presentation of the data and strengthened the underlying message. Specifically:

Major issues:

1) Previous work from this group has shown that MYC combines with TOP2A and TOP1 in a “toposome”. The current work does not provide any context for discriminating the actions and trafficking of a TOP2A-MYC complex versus a toposome.

We agree with the Reviewer. At present, it is technically challenging to discriminate toposome-associated TOP2A from other TOP2A by single molecule tracking given that we do not have the capability to label more than one protein type. In addition, the sparse labeling required for SMT means that typically only 5-10 proteins are fluorescently emitting at any timepoint. Even if both TOP2A and TOP1 could be simultaneously labeled and tracked, the chance that we could visualize two molecules diffusing together is negligible, given there are upwards of 100,000 molecules of each protein in every cell. Nonetheless, we have addressed this point indirectly, using other methods, including those helpfully suggested by the reviewer below.

2) (Incidentally it is unclear to this reviewer if the previously defined toposome is the same as the currently described TOPOsome. If they are the same, then the previous term “toposome” should be retained for simplicity. As the previous term has been used already in reviews and other papers, there is no need to complicate the terminology by introducing a new term.)

We apologize for the confusion. We have changed all mentions to “toposome” as the reviewer suggests.

3) The authors focus on diffusion by TOP2A-MYC complexes and do not address the issue of distinguishing these bimolecular complexes from toposomes. What are the relative abundances of TOP2A-MYC versus toposomes?

We thank the reviewer for this excellent and insightful question. Quantitatively distinguishing bimolecular TOP2A-MYC complexes from topoisomes in live cells is challenging. To address this point, we performed co-immunoprecipitation (co-IP) experiments using lysate from HCT116 cells, which were employed throughout the manuscript. Building on our previous findings³ that TOP1 immunoprecipitates TOP2A only in the presence of MYC (and *vice versa*), thereby forming the “topoisome,” we designed an experimental strategy to estimate the relative proportions of TOP2A in the topoisome *versus* in bimolecular complexes with MYC. Lysates were subjected to two sequential co-IPs: the first using anti-TOP1 antibodies, and the second—applied to the supernatant from the first IP—using anti-MYC antibodies. The immunoprecipitated samples were analyzed by Western blotting together with the input material, and the fraction of TOP2A in each complex was quantified relative to the input (**Extended Data Fig. 4c**). Based on these analyses, we estimate that approximately 10% of TOP2A interacts with MYC as part of the topoisome, in line with our previous estimation in U2OS cells under MYC overexpression³, while 4% forms TOP2A–MYC complexes (**Extended Data Fig. 4d,e**). We note, however, that these values should be regarded as approximate, as they depend on antibody efficiency and IP stringency conditions. These new results are now shown as part of the revised manuscript.

4) If TOP1 were to be doped into the TOP2A condensates *in vitro*, how would the rates of diffusion vary?

Like the Reviewer, we were also curious to determine whether TOP1 changes TOP2A dynamics in relation to MYC. While measuring the diffusion of individual proteins within TOP2A condensates (either *in vitro* or *in cells*) is technically challenging, we approached this request in three different but complementary ways.

Firstly, we generated a recombinant SNAP-tagged TOP1 protein to include in our TOP2A droplet assays and visualize how TOP1 partitions with the TOP2A-MYC condensates. Please note that there have been no prior *in vitro* investigations of TOP1 in the context of condensate formation. Prior to these experiments, we purified and biochemically validated the SNAP-tagged TOP1 protein, confirming that the tag does not alter TOP1 enzymatic activity. We then measured whether TOP1 forms droplets on its own, whether it co-localizes with TOP2A droplets +/- MYC, and whether its addition affects the droplets dynamics. We found that fluorescently labeled TOP1 colocalizes into TOP2A condensates both in the presence or absence of MYC, whereas only a few droplets form with TOP1 + DNA as compared to TOP2A + DNA (**Extended Data Fig. 5h**). Quantification of the average droplet size revealed that MYC induced a decrease in TOP2A droplet size even in the presence of 500 nM TOP1. Thus, MYC’s ability to reduce TOP2A condensate size and sedimentation likely extends to the entire topoisome. These new results are included in the revised **Fig. 4g,i**, and **Extended Data Fig. 5m**

Secondly, following the reviewer's suggestion (see below) we briefly treated cells with camptothecin (CPT)—to trap TOP1 on the DNA and deplete free TOP1 from the topoisome—and then measured TOP2A diffusion by SMT. TOP1 inhibition increased TOP2A diffusion across all fractions (**Extended Data Fig. 4h**). Since the MYC–TOP2A interaction persists in the absence of TOP1, these results suggest that MYC limits complex size to promote TOP2A diffusion. However, because CPT also inhibits transcription⁴, the increased diffusion across all the fractions might be partly a result of transcriptional inhibition. Therefore, we engineered the HCT116 Halo-TOP2A with a TOP1 degron tag to monitor TOP2A diffusion after 2h of auxin-induced TOP1 degradation⁵ (**Extended Data Fig. 4f**) Under these conditions, the diffusion coefficient of TOP2A increased specifically in the slow and bound fractions, while no significant change was observed in the fast fraction. These findings support the notion that in the absence of TOP1, the TOP2A-MYC complex is smaller and that the topoisome likely occurs predominantly in the transcription condensates and on the chromatin. All these new results are now included in the revised **Extended Data Fig. 4f-h**.

5) How would the addition of a TOP1 poison (i.e. CPT) influence the diffusion rate of TOP2A?

We have performed this experiment. Please, see the previous point.

6) Line 283. It is not clear to me that condensates are forming on the ionically immobilized-DNAs rather than polymer encagement TOP2A molecules. The authors assert that because the sizes of spots vary, proves that multiple TOP2As must be included—this assertion doesn't hold water in my view. The sizes of the interstices of the tangled DNA molecules that encage a single TOP2A would be expected to vary considerably, and if so then, so to would the excursions of the trapped molecules vary accordingly. But, if the number of molecules per cage varied, then the brightness would be expected to change incrementally. So, the relative immobility of the TOP2A and its occurrence at very concentrations may not be a result of phase separation but may reflect entrapment within a DNA matrix due to a high local concentration of binding sites.

We thank the reviewer for this thoughtful feedback and for offering an alternative interpretation regarding the nature of the observed TOP2A puncta. We have carefully considered this explanation. However, we respectfully disagree with the interpretation that the observed structures arise from polymer entrapment within a DNA matrix, and we find stronger support for the hypothesis that the TOP2A puncta represent condensates. Our interpretation is based on several independent lines of evidence, as detailed below.

First, if the observed puncta were due solely to DNA entrapment, one would expect to observe DNA aggregates or puncta in the absence of TOP2A. However, when imaging DNA alone under the same conditions, we observe isolated DNA molecules without

clustering. This indicates that DNA itself does not spontaneously form larger clusters/structures and that the presence of TOP2A is required to organize DNA into puncta. In other words, it is TOP2A that recruits and organizes DNA, not *vice versa* (**Rebuttal Figure 1**).

Rebuttal Figure 1 Representative images from experiments with and without TOP2A. The images show the fluorescently labeled DNA without (left) and with (right) TOP2A. The results show how DNA, on its own, is dispersed evenly across the functionalised glass surface, while in the presence of TOP2A it is found in differently sized puncta.

Second, we also detect TOP2A puncta under conditions where DNA is absent. This directly addresses the reviewer's concern: if polymer engagement were the only mechanism, such puncta should not form without DNA. The presence of TOP2A puncta in DNA-free environments supports the idea that self-association or multivalent interactions among TOP2A molecules contribute to puncta formation, consistent with phase separation or percolation (**Rebuttal Figure 2, Extended Data Fig. 5d and Fig 4a, top panel**).

Rebuttal Figure 2 (Extended Data Fig. 5d). Representative image from experiments with TOP2A without DNA. The image showcases how TOP2A spontaneously organizes in puncta.

Finally, regarding the reviewer's expectation of discrete brightness steps, we agree that for an ideal, perfectly labeled and photostable fluorophore this might be observed. However, in practice, fluorophore blinking, photobleaching, and sub-stoichiometric labeling all contribute to variability in signal intensity. These factors are well-known limitations in single-molecule fluorescence and explain why puncta brightness does not appear in discrete increments. We have revised the manuscript to include these clarifications and additional supporting data to make our reasoning more explicit.

7) Line 394. The authors claim that there is no general downregulation of transcription after auxin-depletion of MYC. Therefore, the loss of TOPSAcc's at promoters was not due to decreased transcription. But they make this assessment using whole cell fluorescence after EU incorporation. At the cellular level rRNA accounts for the vast majority of RNA synthesis. So, without some measure of ongoing RNAPII activity, the authors cannot be certain that transcription is unimpaired. In fact, wouldn't ongoing transcription be expected to decline when MYC is degraded?

The reviewer is correct that MYC degradation would ultimately lead to a global reduction in transcription. However, we do not expect this effect to occur within the 90-minute time frame of auxin treatment. Nevertheless, following the reviewer's suggestion, we quantified nascent transcription genome-wide using SLAM-seq, a method based on metabolic labeling of newly synthesized RNA². The data show that transcriptomic profiles are essentially unchanged upon 90min of auxin-induced MYC degradation indicating that the changes in TOP2A activity are attributable to the direct effect of MYC on TOP2A dynamics (**Fig. 5g,h** and **Extended Data Fig. 6g,h**).

Minor issues:

8) Line 84: co-localization, yes, but on what basis is this a condensate?

This assumption is based on recent evidence from the Cissé Lab. Mediator was found to form stable clusters that associate with chromatin, display properties consistent with phase-separated condensates, and are sensitive to 1,6-hexanediol (HD)¹. These clusters interact with Pol II clusters within transcriptional condensates *in vivo*. Since Mediator and TOP2A clusters co-localize and exhibit similar size-distribution, we infer that TOP2A clusters may likewise correspond to transcriptional condensates. If this interpretation is correct, TOP2A clusters should similarly dissolve upon treatment with HD. To test this possibility, we examined (1) the persistence of TOP2A puncta and (2) the dynamics of TOP2A following HD treatment. Results confirm that HD disrupts Mediator-containing transcriptional condensates (**Extended Data Fig. 1d**), in line with previous findings⁶ and, more importantly, reduces the persistence of TOP2A puncta in the nucleoplasm (**Fig.**

1d,e). Consistent with these microscopy results, HD treatment also diminishes the spatial cross-correlation between slow-diffusing and bound TOP2A molecules, with a two-folds reduction relative to the control, indicative of decreased proximity (**Extended Data Fig. 3a-c**). Together with our previous observation that HD induced TOP2A relocation to the nucleolar compartment (also shown in **Extended Data Fig. 1d**), the result provides additional evidence linking TOP2A to Mediator-containing condensates, and that those are likely associated with TOP2A slow fraction (please, see also point **30** of this Response Letter).

However, we acknowledge that these results do not constitute direct evidence. Additional experiments, such as tagging Mediator in HCT116 TOP2A–Halo cells and performing light-sheet imaging, would be required to strengthen this conclusion. Given the limited time frame for resubmission, these experiments fall outside the scope of the current revision. We have accordingly revised the text (lines 84–85) to clarify this point.

9) Line 234: relieve, not relief

We thank the reviewer for the suggestion. We have checked with our native English-speaking colleagues and the correct term should be “relief”.

10) Line 247: The authors have an interesting result upon which they make no comment. Triptolide actually increased TOP2A mobility in all cases. This should be mentioned. Does the inhibition of ERCC3 by the drug inhibit PIC formation or does it release previously recruited TOP2A?

We thank the reviewer for drawing attention to this observation. We did not intend to downplay its relevance; in fact, we view this control experiment as key evidence supporting the direct effect of MYC on TOP2A dynamics, independent of MYC ability to regulate transcription.

Regarding the effect of triptolide on TOP2A mobility, the most straightforward interpretation is that triptolide, by inhibiting the helicase activity of XPB⁷, blocks transcription initiation and elongation, thereby reducing the generation of transcription-induced supercoiling⁸. As a result, TOP2A activity becomes less required, and the protein is less engaged with DNA, leading to an overall increase in its diffusion. Importantly, the absence of a significant change in the diffusion coefficient of the fast fraction upon triptolide treatment (**Extended Data Fig. 4b**), further indicates that the size of the freely diffusing TOP2A complex remains unaltered. We have revised the text to include this interpretation.

11) Line 262 how does this compare with in vivo concentration?

This is an excellent point. Just as the reviewer, we were also interested in comparing the *in vivo* concentration of TOP2A with that used in our droplet assays.

Reports of cellular TOP2A concentration vary widely depending on the cell type and method of measurement. Quantitative mass spectrometry (MS) in HeLa cells calculated that an average cell contained approximately one million monomers of TOP2A equating to approximately 900nM/cell⁹. At the other end of the spectrum, a similar MS experiment in HCT116 cells calculated 188pmol of TOP2A per gram of protein, which equates to approx. 2.3×10^4 monomers of TOP2A, or 20mM/cell¹⁰. It is known that TOP2A varies greatly depending on cell confluency¹¹, which could partially explain the variance. Note that these measurements are per cell, so the nuclear concentration could be 2-3 times higher. Therefore, the predicted physiological nuclear concentration could vary between 40nM-3μM, which agrees well with the concentrations used throughout this manuscript. We have acknowledged in the text that the range of TOP2A concentrations used in this report aligns with the reported physiological concentrations.

Reviewer #2 (Remarks to the Author):

Topoisomerases are essential proteins that resolve DNA supercoiling and entanglements, thereby relieving topological stress. In a previous study, the authors' group showed that the oncoprotein MYC stimulates topoisomerases to remove DNA entanglements generated by oncogenic transcription. In the present study, the authors claim that TOP2A in cells exists in a dynamic equilibrium between sequestration in the nucleolus, substrate searching in transcription condensates, and active engagement on chromatin (three distinct diffusion states). Using single-molecule approaches, they suggest that MYC regulates TOP2A activity by affecting the diffusion state of TOP2A-containing complexes ("TOPOisomes"). Interestingly, MYC accelerated the diffusion of TOP2A in the nucleoplasm, potentially by limiting the size of protein complexes containing TOP2A. This may enhance TOP2A's substrate-searching ability, thereby increasing its activity at transcribed regions. While the authors' story is intriguing, several points are missing to support their main conclusion, and some aspects of data interpretation and presentation are also weak. My specific comments are as follows.

We appreciate the reviewer's thoughtful assessment and recognition of the significance of our study. Many of the reviewer's comments request additional evidence supporting our conclusions. We thank the reviewer for the opportunity to clarify and strengthen these key points in the revised manuscript.

Major points:

12) Fig. 1. As the authors mention, TOP2A expression increases during S-phase, peaks in mitosis, and is subsequently degraded (Refs. 48, 49). The nucleolar accumulation of TOP2A during the cell cycle should be presented.

Thank you for giving us the opportunity to clarify this. To measure TOP2A nucleolar accumulation over the course of the cell cycle, we treated HCT116 cells with EdU prior to fixation to label S-phase cells, and used DAPI nuclear intensity to distinguish G1 and G2 populations, in line with previous reports¹². We then calculated TOP2A nucleolar:nucleoplasmic ratio as performed elsewhere in the manuscript. We did not detect significant differences in overall nucleolar:nucleoplasmic ratio across G1, S, and G2 phase despite a significant increase in TOP2A levels in G2 phase. This suggests that the equilibrium persists independently of TOP2A expression, at least with respect to the cell cycle. These new data are now included in **Extended Data Fig. 1f,g** of the updated manuscript.

In addition, did the total TOP2A level in the nucleus change after either TOP1 depletion or ICRF treatment? It is important to rule out the possibility that the decrease in the nucleolar/nucleoplasmic ratio is due to changed TOP2A levels rather than translocation.

This is a valid point, and we thank the reviewer for raising it. Based on the quantification of TOP2A intensity from the microscopy pictures, we have not found that the overall TOP2A levels in the nucleus change upon either ICRF treatment nor TOP1 degradation. These data are now included in **Extended Data Fig. 2c,d** of the updated manuscript.

13) Fig. 2I–J. Although the authors claim that the slowly diffusing TOP2A corresponds to molecules diffusing in transcriptional condensates, direct evidence appears to be lacking. The data only show that the TOP2A inhibitor ICRF-193 reduces the accumulation of TOP2A at MED4 puncta, whereas the slow fraction remains unchanged following ICRF treatment (Fig. 2C).

We thank the Reviewer for this insightful comment, which is related to a similar point raised by Reviewer #1. The Reviewer is correct that we do not have direct evidence demonstrating that the slow-diffusing fraction of TOP2A corresponds to molecules moving within transcriptional condensates. It is technically unfeasible to address this directly by single-molecule tracking (SMT), as intranuclear SMT is performed in a sparse labelling regime, strongly reducing the probability of observing correlated motion between two differently labelled proteins.

Nevertheless, we observe that the slow fraction of TOP2A displays several features consistent with proteins localized within transcriptional condensates, including (1) retention within a discrete compartment and (2) an average diffusional anisotropy greater than 1, restricted to regions of approximately 150 nm, consistent with the reported dimensions of transcriptional condensates¹³.

In response to the Reviewer's comment, we have also sought to provide additional evidence supporting this hypothesis. Mediator complexes are known to form stable, chromatin-associated clusters that exhibit hallmarks of phase-separated condensates⁶.

Because transcriptional condensates are disrupted by 1,6 hexanediol (HD), we treated HCT116 cells with HD and examined (1) the persistence of TOP2A puncta and (2) the behavior of TOP2A by SMT analysis. The results show that HD disrupts Mediator-containing transcriptional condensates (**Extended Data Fig. 1d**), confirming previous findings⁶ and, more importantly, reduces the persistence of TOP2A puncta in the nucleoplasm (**Fig. 1d,e**). Consistent with these microscopy results, HD treatment also diminishes the cross-correlation between slow-diffusing and bound TOP2A molecules, with a two-folds reduction relative to the control, indicative of decreased proximity. Together with our previous observation that HD induced TOP2A relocation to the nucleolar compartment (also shown in **Extended Data Fig. 1d**), the result provides additional evidence linking TOP2A to Mediator-containing condensates, and that those are likely associated with TOP2A slow fraction (please, see also point **30** of this Response Letter).

To prevent any potential misunderstanding, we have rephrased the sections of the text to clearly state that we lack direct evidence for TOP2A localization within transcriptional condensates. We also note that further experiments, such as tagging Mediator in HCT116 TOP2A–Halo cells and performing light-sheet imaging, would be required to conclusively address this question. However, given the limited time frame for resubmission, such experiments fall outside the scope of the current revision.

Finally, regarding the Reviewer’s observation that “the slow fraction remains unchanged following ICRF treatment (Fig. 2C)”, we would like to clarify that ICRF-193 treatment alters the entire distribution of molecular displacements, resulting in a change of the measured diffusion coefficients for each of the three species (bound, slow diffusing and fast diffusing). The apparent lack of change in the slow fraction might thus depend on such classification into three species. In addition, we now analyzed the SMT data in CTRL and ICRF-193 treated cells using SASPT, a method that allows to compute the full distribution of diffusion coefficients observed (shown in the new **Fig. 2b**). The analysis confirms the presence of three peaks in the spectra of diffusion coefficients, and reveals that upon ICRF-193 the peak corresponding to fast diffusing molecules is slightly shifted towards larger diffusion coefficients (and is largely reduced in size), while slow moving molecules move slightly slower.

14) Fig. 5E. I wonder whether TOP2A can form liquid droplets under the buffer conditions used in the Q-trap assay. The buffer used in the droplet assay contained 20% glycerol, which may be required for TOP2A–DNA droplet formation. The buffer for the Q-trap assay, however, did not contain glycerol. If TOP2A does not form droplets under the Q-trap assay conditions, how can MYC accelerate the binding events of TOP2A at DNA crossovers?

We thank the Reviewer for raising this important point. We would like to clarify that the buffer used for the droplet assay performed on the functionalized substrate (see **Fig. 4d,e** and **Extended Data Fig. 5j**) did **not** contain glycerol, yet TOP2A droplets were still readily detectable under these conditions. Therefore, we have no reason to believe that TOP2A would be unable to form droplets under the Q-trap assay conditions.

15) Overall, the quality of data presentation is low. For instance, in Fig. 2l and Extended Data Fig. 3A, are the scale bars accurate? The TOP2A puncta appear to be around 5 μm in size, which seems too large.

We apologize for the inaccuracy. We realize that our microscope puncta average images were labeled incorrectly. We have now corrected the scale bars to be 200 nm for **Fig. 1c**, **Fig. 2i** and **Extended Data Fig. 3c**, and double-checked all other scale bars.

16) Extended Data Fig. 6g. This figure indicates that the cell population with low EU increased after MYC depletion. This appears to conflict with the authors' interpretation (Line 396).

We observe only a non-significant change in the cell population with low EU levels following MYC depletion. However, since all Reviewers expressed concerns regarding the EU assay, we have re-evaluated transcriptional activity using an orthogonal genomic-based approach. Specifically, we quantified nascent transcription by SLAM-seq, a method based on metabolic labeling of newly synthesized RNA². The results show that transcriptomic profiles are essentially unchanged upon 90min of auxin-induced MYC degradation, nor MYC depletion altered the transcription levels of the top 1500 transcribed genes, indicating that the changes in TOP2A activity are attributable to the direct effect of MYC on TOP2A dynamics (**Fig. 5h** and **Extended Data Fig. 6g,h**).

Minor points:

17) Fig. 1A. This panel is confusing. DAPI staining is needed to visualize the nuclear structure.

We agree with the reviewer that including DAPI staining would be optimal to visualize the nuclear structure. Due to a technical consideration—to avoid DAPI signal bleed-through to other channels—we chose to omit it. This limitation is now explicitly mentioned in the Methods section.

18) Line 95. It is known that RNAPI transcription inhibitors alter the nucleolar structure (nucleolar cap formation).

The Reviewer is correct that RNAPI transcription inhibitors can alter nucleolar structure and induce nucleolar cap formation. However, under our treatment conditions and time points, we did not observe such pronounced changes in fibrillarin staining, a well-

established marker of nucleolar organization. We have rephrased the corresponding text in the manuscript to clarify this point as follows: “under our experimental conditions”.

19) Which MYC subtype did the authors use? Please specify in the Methods section.

We apologize for the inaccuracy. It is the c-MYC isoform. We have included this information in the text.

20) Line 222. Structured illumination light sheet microscopy (SIM)?

Thanks for requesting clarification. We have updated the text to describe the precise method used—multifocus SIM (mSIM)—and referenced the appropriate paper.

21) Code availability: The provided codes (<https://github.com/don-cameron/Cameron-et-al.-2025/tree/main>) have empty “readme.md” files for pixel correlation plotting and for the general instructions for the entire dataset!

We thank the reviewer for pointing this out to us. We have updated the readme files on Github to provide a short explanation of the code.

Reviewer #3 (Remarks to the Author):

Reviewer #3 (Remarks on code availability):

I co-reviewed the code with one of the reviewers who provided the listed reports.

Reviewer #4 (Remarks to the Author):

This is very interesting and strong work built on the previous finding from some co-authors as well as an existing literature. It provides a step forward in the understanding of MYC function which is a big and important question in Multiple fields. Study used an impressive set of biophysical, biochemical and cell-based methods.

Below is my review:

The manuscript of Cameron and co-authors uncover one of the mechanisms of how MYC boosts transcription. They provided data that TOP2A exists in cells in a balanced distribution between nucleolar and nucleoplasm. In nucleoplasm, TOP2A is distributed between different states depending on its diffusion coefficients: a bound fraction interacting or catalytically engaged with chromatin, a slow fraction enriched in transcription condensates, and a fast fraction diffusing through the nucleoplasm. They

then showed that degradation of MYC reduced TOP2A diffusion rates in slow and fast fractions. They showed that MYC promotes TOP2A mobility via reducing the size of TOP2A droplets in cell free conditions and TOP2A complexes in cells. This allows faster search of targets by TOP2A and more efficient resolution of supercoiling needed for transcription.

This is very strong and rigor study using multiple experimental approaches, as well as accurately and clearly written. I just have several questions/comments, which I hope authors can clarify.

We sincerely thank the reviewer for the encouraging and positive assessment of our work. We also appreciate the recognition of the technical challenges and the significance of our findings. Many of the reviewer's comments focus on clarifying aspects of the presented results, and we are grateful for the opportunity to address these points and further strengthen the manuscript.

22) Lines 78-79: "was lost along with nucleolar protein fibrillarin upon further treatment with RNase A (Fig. 1b and Extended Data Fig. 1d), suggesting its association with nucleolar structures was maintained by ribosomal RNA". Was the mechanism of TOP2A association with rRNA already known? Please provide refs or explain.

Thank you for the opportunity to clarify this point. To our current knowledge, there are no reports of TOP2A specifically binding to ribosomal RNA. However, proteome screening has shown human TOP2A binding to mRNA^{14,15} which may suggest a general RNA-binding property of TOP2A. As we mention in the discussion, the rat TOP2A CTD contains a region that is essential for nucleolar localization and also confers sensitivity to RNA-mediated inhibition of TOP2A activity¹⁶, though this has not yet been validated with human TOP2A. Nevertheless, this suggests that high levels of RNA can repress TOP2A activity, which agrees with our general hypothesis that the RNA-rich nucleolus can sequester TOP2A.

Since investigation of the mechanism behind TOP2A's association with the rRNA was somewhat outside the scope of the manuscript, we kept speculation about the role of ribosomal RNA on TOP2A activity and sequestration to the discussion. However, in light of the Reviewer's question, we have included a sentence mentioning this research in the relevant Results section.

23) Contradictory impression between image Fig 1B vs quantitation Fig. S1D. Control image shows signal from nucleolus and nucleoplasm, CSK image – only nucleolar signal, RNase image – only nucleoplasm. Quantitation shows that control = RNase. Please explain.

We thank the reviewer for noticing this inconsistency. We realized that quantification of the Nucleolar:Nucleoplasmic TOP2A ratio is not feasible in the CSK+RNase condition because this treatment removes fibrillarin, which we use to define the nucleolar compartment. As a result, the apparent agreement between Nucleolar:Nucleoplasmic TOP2A ratio in the control and RNase conditions in the previous quantification (old **Extended Data Fig. 1d**) was an artifact of this limitation. To avoid misinterpretation, we have removed the full graph and replaced it with representative images from the four biological replicates (**Fig. 1b** and **Extended Data Fig. 1b**)

24) Figure 1E and F and all other plots of the same type show very few dots while more around thousand cells were measured and mentioned in the legend. What do dots represent?

We thank the reviewer for giving us the opportunity to clarify this point. The plot shows the average of independent replicates, with each experiment representing the mean of approximately 1,000 cells. We have added this information to the figure legend.

25) Lines 123 – 125: “ We observed an increase in DNA damage only after TOP1 degradation by auxin and etoposide treatment” and Fig.1F, ext Fig,2A and B. It is unclear why the gammaH2AX signal increased only after TOP1 degradation. Etoposide interactions with TOP2A should be independent on TOP1.

We thank the reviewer for giving us the opportunity to clarify this point. Because topoisomerases can partially compensate for each other's function, degradation or inhibition of one enzyme (in this case, TOP1) is expected to trigger a compensatory increase in DNA supercoiling. This, in turn, promotes enhanced engagement of TOP2A on DNA to relieve the accumulated torsional stress. Such compensatory behavior has been previously documented by others¹⁷. In our experimental setup, we selectively degrade TOP1 and apply a low dose of etoposide. The observed increase in γ H2AX signal—used here as a proxy for TOP2A activity—occurs only upon combined auxin and etoposide treatment. This suggests that the elevated supercoiling resulting from TOP1 loss is resolved by TOP2A, leading to its increased activity and consequent DNA damage signaling. We have clarified this point better in the text.

26) Also based on presented images – without TOP1 degradation, it is decreased in response to etoposide treatment in ext fig 2A, and increased in ext fig 2b.

We believe the reviewer is referring to the comparison between **Extended Data Fig. 2c** and **Extended Data Fig. 2d**, where γ H2AX staining is shown. We apologize for selecting the wrong image. We have replaced the old **Extended Data Fig. 2c** (new **Extended Data Fig. 2e**) with a more representative example that accurately reflects the underlying data. Please note that a low dose of etoposide was intentionally used in these experiments, as it is not expected to induce detectable DNA damage under normal DNA topology

conditions. Accordingly, the differences observed between CTRL, CTRL + Eto, and auxin treatments are not statistically significant and represent background γ H2AX signal levels.

27) Lines 151-153: “Notably, TOP2B distribution across the nucleus was much more homogeneous as compared to TOP2A, although ICRF-193 also caused a slight re-localization nucleolus-to-nucleoplasm (Extended Data Fig. 2f,g) suggesting it may also be responsive to similar stimuli to TOP2A, although the TOP2A response was more pronounced.” Is there enough structural difference between two topoisomerases to explain why this mechanism is much more pronounced in TOP2A vs TOP2B. It may be good to discuss.

This is an excellent point. Like the reviewer, we are intrigued by the distinct behavior of TOP2B compared with TOP2A. Please note that TOP2B forms a different topoisome (including MYCN and TOP1) as compared to TOP2A (including c-MYC and TOP1)³. Although TOP2A and TOP2B share 77% of overall homology¹⁸, this is largely confined to the N-terminal three quarters of the proteins, which contains the catalytic core. In contrast, their intrinsically disordered C-terminal domains (CTDs) show only ~46% sequence conservation. The CTDs are known to contribute substantially to functional divergence between the two enzymes. For instance, CTDs of TOP2A and TOP2B differently affect their interaction with DNA, catalytic activity¹⁹, their ability to phase separate²⁰, accumulate within the nucleus, and bind RNA²¹. Thus, it is reasonable to speculate that the molecular differences between the CTDs, including their distinct distributions of charged amino acids and isoelectric points²⁰, may underlie the more pronounced nucleolar-to-nucleoplasmic redistribution observed for TOP2A compared with TOP2B. Further experiments in which the CTDs of TOP2A and TOP2B are swapped may be required to directly test this hypothesis. We have included this point in the Discussion.

28) Line 163 “we found that nuclear TOP2A diffusion could be categorized into three distinct populations.” It would be good to show the actual distribution of diffusion coefficients, like histogram or in some other way.

We thank the reviewer for this helpful suggestion. Following this comment, we applied the SASPT algorithm (see Methods section for details) to extract and visualize the full spectrum of diffusion coefficients from the SMT data. The results are consistent with our previous analysis (old **Fig. 2b**), highlighting a broad distribution of diffusion coefficients, with three peaks, corresponding to bound ($\sim 0.01 \mu\text{m}^2/\text{s}$), slow diffusing ($\sim 0.1 \mu\text{m}^2/\text{s}$) and fast diffusing molecules ($\sim 1 \mu\text{m}^2/\text{s}$) and confirming that upon ICRF the fraction of fast-diffusing TOP2A molecules is largely reduced, and that slow-diffusing molecules are further slowed down. The new data are now included in the revised manuscript (new **Fig. 2b**).

29) Figure 2G is unclear. Should it contain colored lines according to the colors of FAST and SLOW? What exactly is the black line?

We thank the reviewer for the opportunity to clarify this point. The black line represents the combined anisotropy values of both the fast and slow fractions. We considered adding colored lines to distinguish the two populations; however, this made the graph visually cluttered. Therefore, we opted to indicate “Combined FAST and SLOW” anisotropy values at the top of the panel and in the figure legend for clarity.

30) Line 198: “If the slow TOP2A fraction reflects TOP2A localized in transcription condensates and the bound fraction corresponds to TOP2A interacting with chromatin,” Should not the slow and bound overlap, since bound should perform its activity within transcriptional condensates.

This is a reasonable comment and below we have highlighted our rationale supporting the idea that slow and bound represent two functionally separate populations of TOP2A molecules.

Our interpretation builds on the pioneering work of the Cissé group showing that transcription condensates only transiently associate with gene loci⁶. In fact, transcriptional burst size and frequency are enhanced when the condensate approaches within $\sim 1 \mu\text{m}$ of the target gene. The authors therefore propose a “three-way kissing” model whereby the condensate interacts transiently with gene locus and regulatory DNA elements to control gene bursting.

Additionally, we would like to clarify that the slow and bound fractions are derived from the vbSPT algorithm that classifies single-molecule tracks based on their diffusion coefficients. By definition, these states are non-overlapping, each molecule at any given time point is assigned to a specific state. Moreover, given the state-transitions analysis (see arrows in **Fig. 2c**), it is notable that slow-to-bound transitions are substantially more frequent than fast-bound transitions, suggesting the TOP2A molecules must pass through the slow state before becoming chromatin-bound.

This observation prompted us to test whether slow and bound fractions have higher likelihood to be found close to each other compared to fast and bound fractions. To probe spatial proximity between bound and diffusing molecules, we performed a spatial “cross-correlation” analysis, as previously done in²². Consistent with our hypothesis, slow diffusing molecules were more likely to be found in proximity of bound molecules than fast diffusing ones. The scale of the observed decay is around 250nm, i.e. it is likely to find slow molecules within 250 nm from bound ones. Treatment with 1,6-hexanediol HD, which disrupts Mediator-containing transcriptional condensates (**Extended Data Fig. 1d**), reduced the cross-correlation between slow-diffusing and bound TOP2A molecules, with

a two-folds reduction relative to the control, indicative of decreased proximity. We have included this analysis in **Extended Data Fig 3a-c**.

Together, these observations suggest that the slow fraction represents TOP2A molecules transiently engaging in the vicinity of chromatin, prior to stable binding, analogous to the dynamic approach of transcriptional condensates toward active gene loci.

31) “transcribed chromatin labelled with H3.3” – H3.3 is not a very specific mark of transcription, since although in general it is enriched in transcribed regions, these transcribed regions are not transcribed all the time, taking into account transcription bursts, and transcription of many genes in a time-dependent manner (e.g., cell cycle). More accurate marks of the presence of transcription can be used, like EU incorporation.

We completely agree with the Reviewer that directly visualizing transcribed DNA loci by labelling newly transcribed RNA with EU would be ideal. However, while optimizing and performing this assay, we identified multiple technical issues with this approach which we feel makes it unsuitable for inclusion in the manuscript.

Specifically, we performed STED microscopy of HCT116 cells treated with ICRF and ActD as described before, but this time including a pulse with EU for 10, 30 and 60 minutes before harvesting. We fluorescently labelled the EU by click chemistry, and stained for TOP2A (**Rebuttal Figure 3**).

Rebuttal Figure 3. Left, Average EU signal at TOP2A puncta from images in untreated cells or upon 5 μ M ICRF-193 treatment for the indicated time. 1389-3671 puncta per condition. **Right,** Average EU enrichment at TOP2A puncta across two independent experiments (EU pulse for 10 minutes).

From these data, we observe that the EU signal only co-localized with TOP2A puncta after 10 minutes, but not 30 minutes. After 60 minutes, the signal is in fact anti-correlated, suggesting that at longer timepoints, EU signal that was initially associated with transcribed loci is processed away from the site of transcription, quickly overwhelming the transcription-associated signal. This suggests that only the 10 minutes treatment would be suitable for labelling transcribed DNA loci.

Encouragingly, the 10 minutes results were in agreement with our H3.3 data, where TOP2A increases after ICRF treatment, but not after ActD treatment. However, due to the extremely low level of labelled EU after only a 10 minute pulse, the resulting signal is very weak and less robust than the H3.3 data. Furthermore, because the EU signal is so faint,

EU puncta cannot be reliably detected, requiring us to instead visualize EU signal at TOP2A puncta. In contrast, the H3.3 staining in the original manuscript produced a clear signal, enabling us to measure TOP2A signal at H3.3 puncta, arguably the more relevant measurement to make.

For these reasons, we have decided not to include these EU results in the manuscript. However, in light of the reviewer's concern, we have reworded the description of H3.3-associated chromatin from "transcribed" to "open" to avoid confusion. We hope that this change satisfactorily addresses the Reviewer's comment.

32) Line 242: "we found that although the proportion of TOP2A found in the fast, slow, and bound fractions were largely unchanged"... do you mean the amount or just the ratio between fractions? Then, was the amount changed?

We apologize for being unclear. This line refers to **Fig. 3b** where there are no significant differences between the CTRL and Auxin conditions with respect to the proportion of tracks classified either as fast, slow or bound. In this instance, the raw amount of tracks is not informative as this is entirely dependent on how many fluorophores are activated during SMT imaging, and not at all to do with any biological differences between the two conditions. To clarify this point, we have changed the wording in the updated manuscript.

33) "MYC mutant containing only residues 200-320 (MYC200-320)" why this form of myc is important? It was mentioned later that it does not bind TOP2A, but it should be better to explain at the first instance.

We thank the reviewer for the opportunity to clarify this point. We used the MYC²⁰⁰⁻³²⁰ as negative control in the droplet assay, as we have previously shown that this mutant is unable to interact with TOP2A in co-IP experiments³. We have clarified this point in the text.

34) Fig.4H "average of six experiments": what four data points represent?

Thanks for pointing this out to us. We apologize for the inaccuracy. It should be "average of four experiments". We have corrected the typos.

35) Why did they use K562 MYC-AID cells in just one section of the manuscript if they already made HCT116 MYC-AID cells?

We thank the reviewer for the opportunity to clarify this point. We chose to work with the K562-MYC degron cells for the gradient experiment because its growth in suspension allows for more rapid and homogeneous cell lysis, besides enabling a more accurate estimation of cell number, a critical factor for the reproducibility of this assay. Moreover, using multiple cell lines ensures that our findings are not cell line-specific, thereby strengthening the generality of the observed trend.

36) Fig. 5A: Was the exposure for the control and auxin the same? It looks like the amount of TOP2A in the auxin condition has been increased. Some of the replicates in the extended data provide a better illustration of the author's interpretation.

Yes, the exposure settings for the control and auxin-treated samples are identical. To improve clarity, we have now moved all western blots from the gradient experiment to **Extended Data Fig. 6**, grouping them together so that the results can be more easily compared across conditions and replicates.

37) Crossover experiment: what was visualized and quantified? Individual TOP2A molecules or droplets.

We thank the Reviewer for raising this point. To determine whether a single dimer or multiple molecules are binding the DNA crossover, we would need to bleach the droplets and count the number of bleaching steps. However, in this instance it is not possible because TOP2A is continually associating and dissociating from the DNA crossover (see trace in **Fig. 5d**). There is also high background signal due to free, fluorescently labeled TOP2A in the solute making it difficult to distinguish individual binding events. Nonetheless, we expect that the majority of the binding events are TOP2A droplets given the variance in peak intensity of TOP2A signal at the crossover (**Fig. 5d**). This is also supported by our new data in **Extended Data Fig. 5d** (described also in comment **6** of this Response Letter), where we can see droplets of multiple TOP2A molecules existing at similar TOP2A concentrations in the absence of DNA. We have edited the text for clarity.

38) Fig. 5H: Does degradation of MYC lead to reduced transcription? Total EU incorporation was not reduced, but expression is a minor fraction of total transcription. How about these 1500 genes?

This point was raised by all the reviewers; therefore we quantified nascent transcripts levels across the genome using SLAM-seq after 90 minutes of auxin-induced MYC degradation. This short-term depletion had essentially no effects on the transcriptional output of the cells (**Extended Data Fig. 6 g,h**), either globally or among the top 1500 most expressed genes (**Fig. 5h**). These findings further strengthen the interpretation that the changes in TOP2A activity result from direct regulation by MYC rather than secondary transcriptional effects (**Fig. 5g**).

39) Line 461-463: "TOP2A molecules decrease by approximately 10%-15%. According to the Stokes-Einstein relationship, in first approximation, diffusion coefficients would scale as the cubic root of the diffusing complex, thus corresponding to roughly 30% to 50% larger complexes upon MYC degradation". Math here is unclear. Cubic root of 30 and 50 are around 3 and 3.6 and not 10-15.

We are grateful to the reviewer for the comment, we have revised the text to make it more clear. The relative fold change of TOP2A diffusion coefficients observed upon MYC degradation is:

$$\frac{D_{noMYC}}{D_{MYC}} = 0.85-0.9$$

According to the Stokes-Einstein relationship, the diffusion coefficient scales as the inverse of the molecule hydrodynamic radius, r . For globular proteins, the radius is, in first approximation, proportional to the cubic root of the molecular weight M . Thus:

$$\frac{M_{noMYC}}{M_{MYC}} \approx \left(\frac{r_{noMYC}}{r_{MYC}}\right)^3 = \left(\frac{D_{MYC}}{D_{noMYC}}\right)^3 = 1.37x \text{ to } 1.62x$$

In other words, TOP2A complexes upon MYC degradation should roughly be 30-50% larger.

References

1. Cho, W.-K. *et al.* Mediator and RNA polymerase II clusters associate in transcription-dependent condensates. *Science* 361, 412–415 (2018).
2. Muhar, M. *et al.* SLAM-seq defines direct gene-regulatory functions of the BRD4-MYC axis. *Science (New York, N.Y.)* 360, 800–805 (2018).
3. Das, S. K. *et al.* MYC assembles and stimulates topoisomerases 1 and 2 in a “topoisome.” *Mol Cell* 82, 140-158.e12 (2022).
4. Pommier, Y. Topoisomerase I inhibitors: camptothecins and beyond. *Nature reviews. Cancer* 6, 789–802 (2006).
5. Wiegard, A. *et al.* Topoisomerase 1 activity during mitotic transcription favors the transition from mitosis to G1. *Mol. Cell* 81, 5007-5024.e9 (2021).
6. Du, M. *et al.* Direct observation of a condensate effect on super-enhancer controlled gene bursting. *Cell* 187, 331-344.e17 (2024).
7. Titov, D. V. *et al.* XPB, a subunit of TFIIH, is a target of the natural product triptolide. *Nat. Chem. Biol.* 7, 182–188 (2011).
8. Chen, F., Gao, X. & Shilatifard, A. Stably paused genes revealed through inhibition of transcription initiation by the TFIIH inhibitor triptolide. *Genes Dev.* 29, 39–47 (2015).
9. Hein, M. Y. *et al.* A Human Interactome in Three Quantitative Dimensions Organized by Stoichiometries and Abundances. *Cell* 163, 712–723 (2015).
10. Wiśniewski, J. R., Koepsell, H., Gizak, A. & Rakus, D. Absolute protein quantification allows differentiation of cell-specific metabolic routes and functions. *PROTEOMICS* 15, 1316–1325 (2015).
11. Padget, K., Pearson, A. & Austin, C. Quantitation of DNA topoisomerase II α and β in human leukaemia cells by immunoblotting. *Leukemia* 14, 1997–2005 (2000).

12. Roukos, V., Pegoraro, G., Voss, T. C. & Misteli, T. Cell cycle staging of individual cells by fluorescence microscopy. *Nature protocols* 10, 334–348 (2015).
13. Muñoz-Gil, G. et al. Stochastic particle unbinding modulates growth dynamics and size of transcription factor condensates in living cells. *Proc. Natl. Acad. Sci.* 119, e2200667119 (2022).
14. Baltz, A. G. et al. The mRNA-Bound Proteome and Its Global Occupancy Profile on Protein-Coding Transcripts. *Mol. Cell* 46, 674–690 (2012).
15. Castello, A. et al. Insights into RNA Biology from an Atlas of Mammalian mRNA-Binding Proteins. *Cell* 149, 1393–1406 (2012).
16. Yasuda, K., Kato, Y., Ikeda, S. & Kawano, S. Regulation of catalytic activity and nucleolar localization of rat DNA topoisomerase II α through its C-terminal domain. *Genes Genet Syst* 95, 291–302 (2020).
17. Miao, Z.-H. et al. Nonclassic Functions of Human Topoisomerase I: Genome-Wide and Pharmacologic Analyses. *Cancer Res.* 67, 8752–8761 (2007).
18. Austin, C. A., Sng, J.-H., Patel, S. & Fisher, L. M. Novel HeLa topoisomerase II is the II β isoform: complete coding sequence and homology with other type II topoisomerases. *Biochim. Biophys. Acta (BBA) - Gene Struct. Expr.* 1172, 283–291 (1993).
19. Broeck, A. V. et al. Structural basis for allosteric regulation of Human Topoisomerase II α . *Nat Commun* 12, 2962 (2021).
20. Jeong, J., Lee, J. H., Carcamo, C. C., Parker, M. W. & Berger, J. M. DNA-Stimulated Liquid-Liquid phase separation by eukaryotic topoisomerase ii modulates catalytic function. *Elife* 11, e81786 (2022).
21. Onoda, A. et al. Nuclear dynamics of topoisomerase II β reflects its catalytic activity that is regulated by binding of RNA to the C-terminal domain. *Nucleic acids research* 42, 9005–9020 (2014).
22. Mazzocca, M. et al. Chromatin organization drives the search mechanism of nuclear factors. *Nat. Commun.* 14, 6433 (2023).

We are pleased that reviewers 1 and 4 expressed appreciation for our revised manuscript and found it to be appropriate for publication in *Nature Communications*. We have carefully addressed the remaining concerns of reviewer 2 as outlined below:

1. Extended Data Fig. 3a–c: I wondered whether these data include the slow and bound particles localized in nucleoli, since most of the slow/bound particles were located in nucleoli, which are not linked to transcriptional condensates.

We thank the reviewer for pointing this out. The cross-correlation analysis was performed exclusively on the nucleoplasmic fast and slow fractions of TOP2A. We have now explicitly clarified this in the figure legend.

2. Fig. 2a: Representative movies of the single-particle tracking data should be presented to aid the readers' understanding.

We thank the reviewer for this suggestion. We have now included two representative movies of the SMT data (Supplementary video 1 and 2).